# High figure-of-merit for ZnO nanostructures by interfacing lowly-oxidized graphene quantum dots

Myungwoo Choi[1,9], Juyoung An[2,9], Hyejeong Lee[3], Hanhwi Jang [2], Ji Hong Park[2,4], Donghwi Cho [5,6], Jae Yong Song[7], Seung Min Kim [4], Min-Wook Oh [8], Hosun Shin [3] ✉ & Seokwoo Jeon [1] ✉

Thermoelectric technology has potential for converting waste heat into electricity. Although traditional thermoelectric materials exhibit extremely high thermoelectric performances, their scarcity and toxicity limit their applications. Zinc oxide (ZnO) emerges as a promising alternative owing to its high thermal stability and relatively high Seebeck coefficient, while also being earth-abundant and nontoxic. However, its high thermal conductivity (>40 W m$^{-1}$K$^{-1}$) remains a challenge. In this study, we use a multi-step strategy to achieve a significantly high dimensionless figure-of-merit ($zT$) value of approximately 0.486 at 580 K (estimated value) by interfacing graphene quantum dots with 3D nanostructured ZnO. Here, we show the fabrication of graphene quantum dots interfaced 3D ZnO, yielding the highest $zT$ value ever reported for ZnO counterparts; specifically, our experimental results indicate that the fabricated 3D GQD@ZnO exhibited a significantly low thermal conductivity of 0.785 W m$^{-1}$K$^{-1}$ (estimated value) and a remarkably high Seebeck coefficient of −556 µV K$^{-1}$ at 580 K.

With the ever-increasing energy demand and consumption of fossil fuels, the researchers are actively focusing on renewable energy technologies[1]. Among the various renewable energy technologies, the thermoelectric system is gaining increased attention because more than 65% of the globally produced energy from industrial machinery is lost as waste heat. Much of waste heat can be directly converted into valuable electrical power via the Seebeck effect[2,3]. The efficiency of thermoelectric systems is evaluated using the dimensionless figure-of-merit ($zT = \sigma S^2 T \kappa^{-1}$) depending on intrinsic material parameters— electrical conductivity ($\sigma$), Seebeck coefficient ($S$), thermal

conductivity ($\kappa$), and absolute temperature ($T$)[4]. Over the past decades, various thermoelectric materials including metal tellurides (e.g., Bi$_2$Te$_3$[5], PbTe[6], etc), metal silicides (e.g., Mg$_4$Si$_7$[7], Mg$_2$Si-Mg$_2$Sn[8], etc), and Si-Ge alloys[9] with high $zT$ values (>1.5) have been actively reported. Although these materials bring thermoelectric technology closer to commercialization, their toxicity, scarcity, and chemical instability remain open challenges. In this regard, metal oxides are potential alternatives because of their nontoxicity, earth-abundance, and high thermal stability, thereby opening up more possibilities for the commercialization of thermoelectric technology[10]. Various metal oxides,

[1]Department of Materials Science and Engineering, Korea University, Seoul 02841, Republic of Korea. [2]Department of Materials Science and Engineering, Korea Advanced Institute of Science and Technology, Daejeon 34141, Republic of Korea. [3]Division of Chemical and Material Metrology, Korea Research Institute of Standards and Science, Daejeon 34113, Republic of Korea. [4]Institute of Advanced Composite Materials, Korea Institute of Science and Technology, Jeonbuk 55324, Republic of Korea. [5]Thin Film Materials Research Center, Korea Research Institute of Chemical Technology, Daejeon 34114, Republic of Korea. [6]Advanced Materials and Chemical Engineering, University of Science and Technology, Daejeon 34113, Republic of Korea. [7]Department of Semiconductor Engineering, Pohang University of Science and Technology, Pohang 37673, Republic of Korea. [8]Department of Materials Science and Engineering, Hanbat National University, Daejeon 34158, Republic of Korea. [9]These authors contributed equally: Myungwoo Choi, Juyoung An. ✉e-mail: hshin@kriss.re.kr; jeon39@korea.ac.kr

such as $SrTiO_3$, $Ca_3Co_4O_9$, and $Na_xCo_2O_4$, have been developed to achieve competitive $zT$ values with conventional thermoelectric materials, as summarized in Supplementary Table 1. However, the synthesis techniques for metal oxides with high thermoelectric performances are quite challenging due to their complex chemical configurations and crystal structures.

Zinc oxide (ZnO), a well-known n-type thermoelectric metal oxide, has emerged as a promising candidate due to its relatively high $S$ ($-100$ to $-400\,\mu V\,K^{-1}$), wide working temperature (570–1500 K), and facile processability compared to other metal oxides[11,12]. Despite the increase in the academic and industrial efforts to realize commercial-scale thermoelectric performances using ZnO, relatively high $\kappa$ ($>40\,W\,m^{-1}\,K^{-1}$) of ZnO remains a critical obstacle and therefore hardly avoids the low $zT$ value ($-1.4 \times 10^{-4}$ at 350 K and ~0.02 at 1000 K)[13,14]. To address this problem, nanostructuring—such as nano-arrays[15], nano-porous materials[16], nanocomposite structures[17], and grain boundary engineering[18–20]—has been proposed to significantly reduce $\kappa$ by scattering of phonons at countless interfaces of nanostructures. Notwithstanding advances in this direction, the resulting random-form factors cause inevitable $\sigma$ degradation or uncontrollable electron scattering. Therefore, it is crucial to design ZnO with uniform nanoscale features to effectively reduce $\kappa$ while maintaining electrical conduction pathways. In this regard, we previously demonstrated a rational design and fabrication of highly ordered three-dimensional (3D) thin-shell ZnO using an advanced lithographic technique, namely, proximity field nanopatterning (PnP)[21–26]. The design achieved a significant reduction in $\kappa$ without degrading $\sigma$ owing to the continuous network, by considering the difference in the phonon and electron scattering lengths[27–29]. Furthermore, the fabricated 3D ZnO has the potential for flexible thermoelectric applications, overcoming the intrinsic brittle nature of ZnO through structural engineering[30].

In addition to structural engineering, interface engineering by incorporating nanosized materials into nanostructured supports is regarded as an efficient strategy, because interfacing not only mitigates phonon propagation but also selectively filters out low-energy electrons to enhance $S$ based on the well-established energy filtering effect[31]. For example, inserting nanocarbon additives (e.g., graphene and carbon nanotube) into nanostructured thermoelectric materials effectively reduces their $\kappa$ owing to nanoprecipitate phonon scattering and increases $S$ through the filtering of low-energy electrons at additive/host interfaces[32–35]. Among nanocarbon materials, graphene quantum dots (GQDs) have an advantage in terms of effective interfacing over graphene or carbon nanotubes because of a sufficient number of active sites to uniformly cover the nanostructured supports. However, a well-matched band alignment with the host metal oxides is essential to form a proper energy barrier at the interface to maximize thermoelectric performance through the interfacing of GQDs. In this sense, lowly-oxidized GQDs fabricated from graphite intercalation compounds (GICs) can be a promising additive for host metal oxides owing to their well-controlled energy states from subdomains that only appear within GQDs under controlled oxidation, compared with defect-rich GQDs synthesized using Hummers' method[36–38]. Previously, we demonstrated the effectiveness of lowly-oxidized GQDs as an interfacing additive to various metal oxides (e.g., $TiO_2$ and $SnO_2$) in photocatalyst and gas sensor applications[39–41]. The interface of GQD/metal oxides can enhance $S$ by filtering out low-energy electrons while lowering $\kappa$ by scattering of phonons at a well-controlled interfacial energy barrier. Therefore, a carefully engineered GQD/metal oxide interface and a uniform 3D nanostructure improve its thermoelectric performance.

In this work, we present a multi-step rational strategy for achieving high thermoelectric performance by interfacing lowly-oxidized GQD with 3D nanostructured ZnO (3D GQD@ZnO). The highly periodic 3D thin-shell ZnO considerably reduces $\kappa$ by effective scattering of long mean free path (MFP) phonons at nanostructured surfaces, which contribute most to the thermal conduction of ZnO at the optimized thickness of 70 nm of the 3D ZnO shell. In addition to the structural factor, grain boundary engineering in the thin-shell ZnO through careful thermal treatment at 250 °C for 2 h induces further phonon scattering and low-energy electrons filtering at grain boundaries, resulting in a slightly improved $zT$ value at an optimum grain size (~30 nm) of thin-shell ZnO. Furthermore, interfacing lowly-oxidized GQDs with the 3D ZnO provides numerous heterogeneous interfaces of GQD/ZnO, forming an interfacial energy barrier of 0.63 eV owing to the energy offset between the lowest unoccupied molecular orbital (LUMO) of GQD and conduction band minima (CBM) of ZnO. This study indicates that the interfacial energy barrier is the key factor for maximizing the $zT$ value by altering electron and phonon transport. Specifically, the low-energy electrons of ZnO, corresponding to energies below the LUMO level of GQD, become trapped at the GQD/ZnO interfacial energy barrier. This electron filtering—allowing only the transport of high-energy electrons above the LUMO level of GQDs—enhances $S$, resulting in a 74% enhancement of power factor ($\sigma S^2$) when compared with that of the bare 3D ZnO despite a slight decrease in $\sigma$ owing to a reduced charge carrier concentration. This interfacial energy barrier also leads to the significant reduction in $\kappa$ from 1.49 to $0.785\,W\,m^{-1}\,K^{-1}$ through the nanoprecipitate scattering of phonons with relatively short MFP. Consequently, the 3D GQD@ZnO heterostructure exhibits an exceptionally high $zT$ value of 0.486 at 580 K, yielding a record high value among ZnO-based materials.

## Results

### Realization of 3D GQD@ZnO heterostructure

Figure 1a shows the fabrication scheme for the 3D GQD@ZnO heterostructure and a scanning electron microscopy (SEM) image of the 3D thin-shell ZnO. First, an advanced 3D lithographic technique, PnP, was used to produce a highly periodic 3D nanostructure with regular periodicity, large surface area, and large-scale uniformity. Thereafter, an aqueous solution of GQDs was prepared to interface with the 3D ZnO support. Because of the hydrophilic surface and 3D interconnected nanonetwork of 3D ZnO, GQDs can be uniformly coated along the surface of the entire structure, enabling uniform and reliable interfacing of the GQDs with the surface of the 3D ZnO support (see the "Methods" section for details). Unlike the conventional functionalization approach for forming chemical bonds between GQD and metal oxides[40], our solution process involved only the physical attachment of GQDs to the ZnO surface to form an interfacial energy barrier, avoiding uncontrolled charge behavior caused by chemical bonds.

The 3D ZnO with many nanostructured surfaces contributes to powerful phonon scattering, resulting in a reduction in $\kappa$[27]. In addition to the nanostructured surfaces of 3D ZnO, as shown in Fig. 1b, the grain boundaries of 3D ZnO and heterogeneous interfaces with GQDs can modify the transport behavior of both phonons and electrons. For a clear observation of the 3D GQD@ZnO heterostructure, high-resolution transmission electron microscopy (HR-TEM) images were obtained (Fig. 1c). An atomic (0002) plane of the wurtzite ZnO structure ($d_{ZnO}$ ~0.26 nm, white circles in Fast Fourier-Transform (FFT) patterns) was described. Furthermore, 5 nm GQDs with a hexagonal arrangement ($d_{GQD}$ ~0.21 nm, yellow circles in FFT patterns) were successfully interfaced on the 3D ZnO surface, as shown in the Fig. 1c. The key factors in improving the thermoelectric properties of 3D GQD@ZnO are the grain boundaries and heterogeneous interfaces, which offer both phonon scattering and low-energy electron filtering. First, phonon and electron transport are influenced by the structural differences at the intrinsic grain boundaries, which provide dual sites for phonon scattering and low-energy electron filtering. Next, for the 3D GQD@ZnO heterostructure with heterogeneous interfaces, an interfacial energy barrier can provide additional dual-function sites for phonon scattering and low-energy electron filtering, because of the

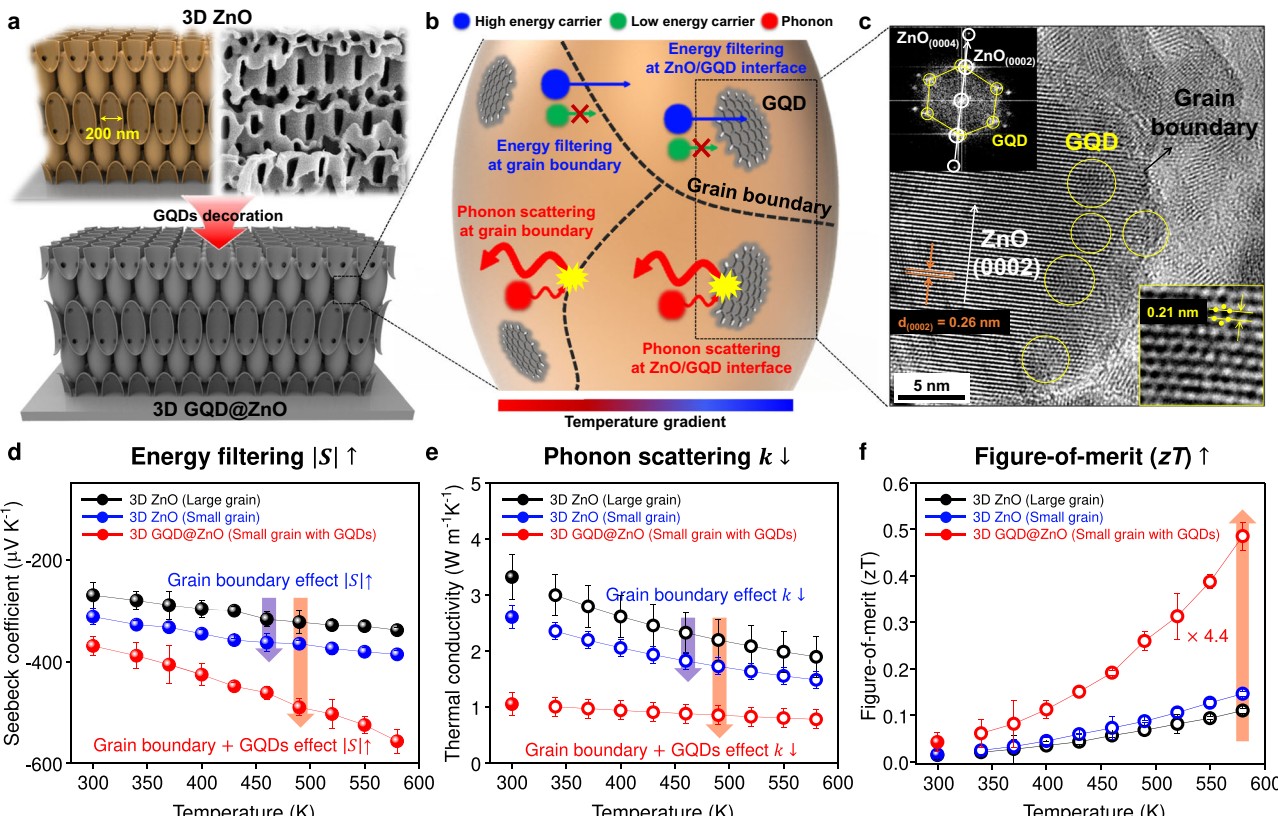

**Fig. 1 | Strategy to improve thermoelectric performance via grain boundary and interface engineering. a** Illustration of graphene quantum dots (GQDs)-decorated 3D ZnO (3D GQD@ZnO) obtained by dipping the 3D ZnO into GQD solution. **b** Schematic of energy filtering and phonon scattering effect at grain boundaries and GQD/ZnO interfaces in the 3D GQD@ZnO. **c** Transmission electron microscopy (TEM) image showing decoration of GQDs in 3D ZnO. Yellow circles of Fast Fourier-Transform (FFT) patterns indicate a hexagonal pattern of GQD, and white circles show an atomic plane of wurtzite ZnO structure. The inset of **c** indicates a high-resolution (HR)-TEM image of GQD coated on 3D ZnO. Temperature-dependent **d** Seebeck coefficient, **e** thermal conductivity, and **f** Figure-of-merit (zT) for 3D ZnO with varying grain sizes and its heterostructure with GQDs (3D GQD@ZnO). Filled circles are measured values, and open circles indicate the estimated values obtained using a modified Debye-Callaway model. Error bars shown in the Fig. 1 represent the standard deviation (SD).

careful design of the energy-band alignment between heterogeneous materials; consequently, it improves the zT value through synergetic effects derived from interface engineering.

To confirm the impact of interface engineering on thermoelectric performance, we systematically evaluated temperature-dependent $S$ and $\kappa$ (Fig. 1d, e). When the grain size of 3D ZnO decreased, the value of $S$ increased and that of $\kappa$ decreased owing to the increase in grain boundary density. Furthermore, as predicted, 3D GQD@ZnO with many heterogeneous interfaces after interfacing GQDs exhibits a significant enhancement of zT value coming from the more increased $S$ and more decreased $\kappa$ (Fig. 1d–f). It was confirmed that the zT value of the 3D GQD@ZnO (0.486) was approximately 4.4 times higher than that of the bare 3D ZnO with a large-grain size at 580 K (Fig. 1f). This observation suggests that an enormous improvement in thermoelectric performance can be achieved through interface engineering in conjunction with structural engineering.

## Fabrication of 3D thin-shell ZnO and grain boundary engineering

A photograph and cross-sectional SEM images were obtained to investigate the structure of pristine 3D ZnO on a sapphire substrate (Fig. 2a, b). First, a polymeric 3D template was fabricated using a PnP technique. The polymeric 3D nanostructure served as a template for material conversion to metal oxide through the atomic layer deposition (ALD) technique for uniform deposition of thermoelectric materials. The plasma etching process was used to remove the polymeric template. The fabrication process for the 3D ZnO is shown in

Supplementary Fig. 1. The body-centered tetragonal symmetry with a periodicity of 600 nm of the 3D thin-shell ZnO is clearly described. To optimize the shell-thickness of the 3D ZnO with respect to phonon transport, we prepared three types of 3D ZnO systems with shell thicknesses of 30, 50, and 70 nm by adjusting the deposition cycles of the ALD process. The average pore sizes for three types of 3D ZnO systems were 280, 230, and 180 nm, respectively, which depend on the shell-thickness of ZnO (Supplementary Fig. 2a, b). The reduction ratios of the thermal conductivity achieved by phonon scattering in 3D ZnO with different shell thicknesses are expected to be similar, given the comparable contribution of thermal conductivity by phonons with MFPs corresponding to different pore sizes[42]. However, when the ZnO shell-thickness exceeds 100 nm, the thermal conduction achieved by phonons with a short MFP, which are transported within the relatively thick shell (100 nm or greater), becomes more significant than the phonon scattering at the nanostructured surfaces[42]. Meanwhile, as the shell-thickness increases, the 3D ZnO nanostructure becomes more uniform and exhibits high structural stability (Supplementary Fig. 2c). For the 30 nm sample with a relatively thin shell, structural stability was relatively lower due to a slight collapse during the removal of the polymer template after ZnO deposition. Therefore, the thin-shell-thickness was optimized at 70 nm, which is sufficient to induce effective scattering of phonons with a long MFP at nanostructured surfaces, along with the structural stability of highly porous nanostructures at high temperatures[27]. Fig. 2c shows a magnified cross-sectional TEM image of the 3D ZnO (shell-thickness ~70 nm), which was obtained by focused ion beam (FIB) milling. The highly periodic interconnected

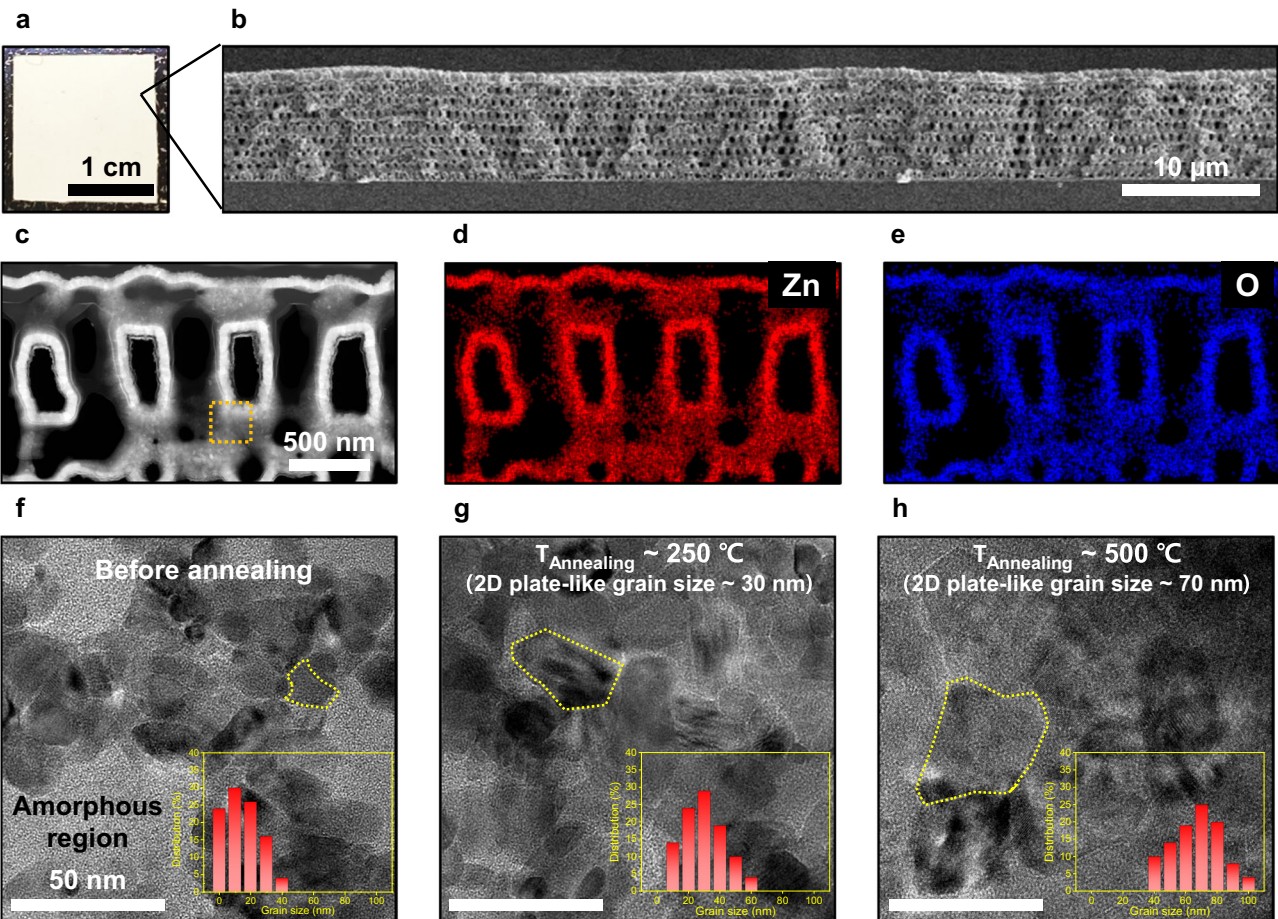

**Fig. 2 | Structural investigation of 3D thin-shell ZnO with varying 2D plate-like grain sizes in thin-shell. a** Photograph of 3D ZnO film on the sapphire substrate. **b** Cross-section of scanning electron microscopy (SEM) image of 3D ZnO. **c** TEM image of 3D ZnO. **d**, **e** Energy dispersive spectrometer (EDS) elemental mapping images corresponding to TEM image of **c**. **f**–**h** TEM images of 3D ZnO samples based on annealing temperature. The insets of **f**–**h** indicate the distribution of 2D plate-like grain sizes on the surface of thin-shell ZnO corresponding to the TEM images of **f**–**h**.

network of the 3D ZnO showed that material conversion from the polymeric template was successfully accomplished without structural degradation after template removal. In addition, elemental mapping obtained from an energy dispersive spectrometer (EDS) indicated that the average atomic compositions of Zn and O had a one-to-one correspondence, demonstrating simultaneous process reproducibility with the structural stability of the ALD-deposited nanostructures (Fig. 2d, e). With respect to the thermoelectric transport behavior, this unique nanostructure can selectively enhance phonon scattering at many nanostructured surfaces without the degradation of electrical properties owing to the difference in MFP (phonon > electron)[43].

In addition to structural engineering, grain boundary engineering is an important factor that can modify the phonon/electron transport[32,44]. Considering the complex but controllable thermoelectric behavior of the 3D ZnO, not only the nanostructured interface of the 3D ZnO but also the grain boundary of ZnO must be appropriately designed. Thus, we performed grain boundary engineering of ZnO through annealing-control experiments. Figure 2f–h shows the HR-TEM images of the surface of the 3D ZnO for various annealing temperatures. The insets in Fig. 2f–h indicate the distribution of the grain size of the 3D ZnO corresponding to the TEM images in Fig. 2f–h. Note that the grain size refers to the size of the 2D plate-like grains in thin-shell ZnO, not the actual 3D grain size. As-fabricated pristine 3D ZnO before annealing exhibited a small grain size (~10 nm) and amorphous region owing to extremely low temperature (90 °C) for ALD. As the annealing temperature increased, the grain size increased

owing to temperature-dependent grain growth behavior, as confirmed by X-ray diffraction (XRD) analysis (Supplementary Fig. 3). The average grain sizes of the 3D ZnO were 30 and 70 nm at annealing temperatures of 250 and 500 °C, respectively. Considering structural stability, we controlled the annealing temperature up to 500 °C because collapse can occur with heat treatment over 500 °C owing to the relatively high porosity (~80%) of the nanostructure.

**Thermoelectric properties of 3D ZnO with varying grain sizes**

To investigate the impact of the grain boundary on the electron/phonon transport behavior, thermoelectric properties were systematically evaluated. In general, an increase in the grain boundary fraction correlated with an increase in the electron and phonon scattering sites. In other words, it can induce an increase in $S$ via low-energy electron filtering while suppressing $\kappa$ through phonon scattering at grain boundaries. We obtained this result experimentally by measuring the thermoelectric properties of the 3D ZnO of various grain sizes. For a thorough understanding of the impact of grain boundary engineering on thermoelectric properties before annealing, small (30 nm, annealed at 250 °C) and large grain (70 nm, annealed at 500 °C) 3D ZnO samples were investigated. Figure 3a, b presents the $S$ and $\sigma$ values of the 3D ZnO samples as functions of temperature (300–580 K). The 3D ZnO showed a negative value of $S$, suggesting that electrons were the majority carriers[12]. Also, an increasing trend in $S$ and $\sigma$ based on the increase in temperature indicated typical characteristics of a nondegenerate semiconductor. Although all samples showed a relatively

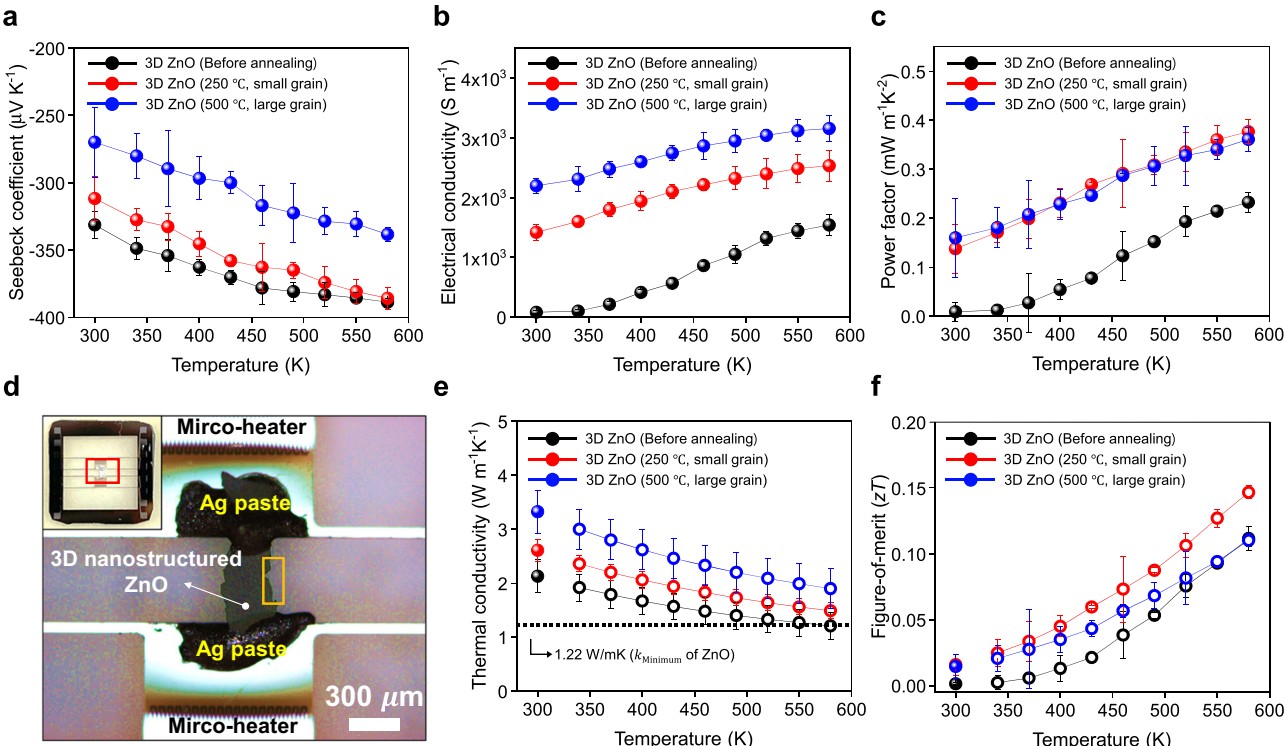

**Fig. 3 | Analysis of electrical/thermal transport properties of 3D ZnO with varying grain sizes.** Temperature-dependent **a** Seebeck coefficient, **b** electrical conductivity, and **c** power factor of 3D ZnO with varying grain sizes. **d** Optical microscopy (OM) and digital (inset) images of 3D ZnO film transferred onto a microfabricated thermoelectric measurement platform (MTMP). **e** Comparison of thermal conductivities of 3D ZnO samples measured on MTMP as a function of temperature. Filled circles are measured values, and open circles indicate estimated values obtained using a modified Debye-Callaway model. **f** Comparison of $zT$ values of 3D ZnO samples. Error bars shown in Fig. 3 represent the SD.

high $S$ originating from semiconductor characteristics with a wide band gap[12], $S$ significantly varied with the grain size. Before annealing, the 3D ZnO exhibited the highest $S$ ($-331\,\mu V\,K^{-1}$ at 300 K) and the lowest $\sigma$ (82 S m$^{-1}$ at 300 K). However, the 3D ZnO with large grains showed contradictory results ($-269\,\mu V\,K^{-1}$ and 2196 S m$^{-1}$ at 300 K), suggesting that the grain boundary serves as a scattering site for charge carriers and facilitates effective energy filtering. The maximum $S$ and $\sigma$ values ($-388\,\mu V\,K^{-1}$ and 1540 S m$^{-1}$ (before annealing), $-385\,\mu V\,K^{-1}$ and 2531 S m$^{-1}$ (for small grain), and $-338\,\mu V\,K^{-1}$ and 3154 S m$^{-1}$ (for large grain)) were achieved at 580 K within the measurement temperature range, implying that the optimal working temperature of the 3D ZnO for the best performance is in the mid-temperature range. The values of power factor ($\sigma S^2$) of the 3D ZnO samples reached 0.233 (before annealing), 0.377 (for small grain), and 0.361 mW m$^{-1}$K$^{-2}$ (for large grain) at 580 K (Fig. 3c), indicating a 62% increase after annealing at 250 °C when compared with that of the as-deposited 3D ZnO by ALD without annealing.

Next, $\kappa$ measurements were performed to investigate the impact of grain boundary on thermal transport. We used a microfabricated thermoelectric measurement platform (MTMP) to measure the in-plane $\kappa$ of the 3D ZnO at 300 K[18,19,27,28,45]. The freestanding 3D ZnO was released from the substrate and transferred onto the MTMP, which could directly measure the $\kappa$ based on the Fourier law (Fig. 3d and Supplementary Fig. 4). Before annealing, the measured $\kappa$ value of 2.13 W m$^{-1}$K$^{-1}$ at 300 K was obtained, which gradually increased to 3.32 W m$^{-1}$K$^{-1}$ for the large-grain 3D ZnO (filled circles in Fig. 3e). This result suggests enhanced phonon scattering with an increase in the grain boundary fraction, especially for phonons with an MFP in the range of several tens of nm. All the 3D ZnO samples showed a significantly reduced $\kappa$ when compared with that of the ZnO bulk materials (>40 W m$^{-1}$K$^{-1}$), implying the significantly enhanced scattering of

phonons, which have an MFP in the range of several hundreds of nm, at numerous 3D nanostructured interfaces.

Owing to the limitation of the $\kappa$ measurements at high temperatures in the MTMP setup, a modified Debye-Callaway model based on the measured lattice $\kappa$ at 300 K and electronic $\kappa$ derived from the Wiedemann-Franz law was used to estimate the temperature-dependent $\kappa$ of the 3D ZnO. In addition, the porosity of the unique 3D thin-shell structure was considered based on the effective medium theory (see the "Methods" section for details). The open circles in Fig. 3e show the temperature-dependent $\kappa$ of 3D ZnO with varying grain sizes in the 340–580 K range. At 580 K, the estimated $\kappa$ values of the 3D ZnO samples were 1.21 W m$^{-1}$K$^{-1}$ (before annealing), 1.49 W m$^{-1}$K$^{-1}$ (for small grain), and 1.90 W m$^{-1}$K$^{-1}$ (for large grain). Especially, the $\kappa$ value of before-annealed 3D ZnO was comparable to the theoretically minimum value of ZnO (1.22 W m$^{-1}$K$^{-1}$, dotted line in Fig. 3e) calculated from the Cahill model[46]. This result validates that extreme phonon scattering occurs by simultaneous use of 3D nanostructuring and grain boundary engineering. In short, thermoelectric $zT$ values of the 3D ZnO with varying grain sizes were calculated by a collective set of the power factor, $\kappa$, and $T$ (Fig. 3f). The 3D ZnO with small grains showed the maximum $zT$ value (0.147 at 580 K) among all the samples (Supplementary Table 2), suggesting the importance of optimizing the thermoelectric parameters with a trade-off relationship.

## Interface engineering of 3D ZnO with GQDs

To uniformly incorporate GQDs into the 3D ZnO with a well-established thermoelectric support through grain boundary engineering, we used the dip-coating method. First, lowly-oxidized GQDs were prepared from GICs via a previously reported method[36] (Supplementary Fig. 5; see the "Methods" section for details). After

dip-coating the 3D ZnO into a GQDs solution, the GQDs were uniformly attached to the surface of the 3D ZnO (3D GQD@ZnO), supported by a vivid color change from whitish to brownish after interfacing the GQDs (Fig. 4a). A TEM analysis was used to confirm the uniform interfacing of the GQDs with the 3D ZnO (Fig. 4b–f). As shown in Fig. 4b, c, the 3D ZnO with a shell-thickness of 70 nm was confirmed. From the magnified TEM images and FFT patterns in Fig. 4d–f, it is also evident that the 5 nm GQDs are uniformly coated on the 3D ZnO structure because of the hydrophilic surface of ZnO and the highly retained sp² carbon arrangement of the GIC-based GQDs, enabling the uniform coating of GQDs in water along the surface of the 3D thin-shell ZnO structure.

X-ray photoelectron spectroscopy (XPS) and Fourier-transform infrared spectroscopy (FT-IR) analyses were performed to further investigate the surface chemical states and elemental composition of 3D GQD@ZnO. As is evident from the XPS survey results (Supplementary Fig. 6a), a carbon peak at 284 eV (C1s peak) was observed in the 3D GQD@ZnO because of the carbon content in the GQDs compared to the pristine 3D ZnO. In addition, the FT-IR results (Supplementary Fig. 6b) indicate the presence of –OH, C = C, COOH, and C-O groups in 3D GQD@ZnO as against only the Zn-O group in pristine 3D ZnO, clearly demonstrating the attachment of GQDs to 3D ZnO. For a more direct observation of the nature of the attachment of the GQDs to the 3D ZnO surface, the C1s and O1s peaks from the XPS results were deconvoluted. Compared with pristine GQDs (Supplementary Fig. 7), no additional peak with a chemical bond was observed in the 3D GQD@ZnO, except for a Zn-O bond (Supplementary Fig. 8), indicating that the GQDs were physically attached to the 3D ZnO without chemical bonding between GQD and ZnO. The attachment of GQDs to 3D ZnO was also confirmed by analyzing the UV−Vis absorption spectra (Supplementary Fig. 9). The 3D ZnO exhibited significantly enhanced absorption

in the UV range when compared with thin-film ZnO, owing to its improved surface area and unique light-scattering effect[25,40]. Moreover, when GQDs were directly attached to pristine 3D ZnO, a slight absorption enhancement was observed throughout the UV and visible light ranges, suggesting a light-activated charge transfer between GQD and ZnO.

For a detailed analysis of the energy-band structure at the GQD/ZnO interface for the 3D GQD@ZnO heterostructure, UV−Vis absorption spectra were obtained for the unstructured ZnO thin films and GQD (Fig. 4g). First, the energy-band gaps ($E_g$) of ZnO and GQD were calculated using the Tauc plot derived from the absorption spectra (inset of Fig. 4g). The 5 nm GQD with discrete $E_g$ ($E_{g\ (GQD)}$ ~ 3.2 eV) indicated a larger $E_g$ than that of ZnO ($E_{g\ (ZnO)}$ ~ 3 eV), which can form an interfacial energy barrier depending on the valence band maximum (VBM) of ZnO and the highest occupied molecular orbital (HOMO) of GQD. Next, we obtained the VBM of ZnO and the HOMO of GQD using ultraviolet photoelectron spectroscopy (UPS) (Fig. 4h). The UPS results determined the low-binding-energy tail ($E_{onset}$) and high-binding-energy cutoff ($E_{cutoff}$) of ZnO and GQD, respectively. The energy levels of the VBM for ZnO and the HOMO for the GQD were calculated using the following equation[47]:

$$-E_{VBM}(E_{HOMO}) = \hbar\nu - (E_{cutoff} - E_{onset})$$

where $\hbar\nu$ = 21.22 eV is the incident photon energy using the HeI excitation source. Based on the calculated $E_g$ of ZnO and GQD, we verified that the LUMO of GQD is located at a higher energy level than the CBM of ZnO (Fig. 4i), resulting in a formation of an energy barrier of 0.63 eV at the GQD/ZnO interface. Therefore, the interfacial energy barrier formed at the heterogeneous interface of the 3D GQD@ZnO can modify the electron/phonon transport behavior, thereby acting as a low-energy electron filtering and phonon scattering site. Notably, the utilization of lowly-oxidized GQDs with a controlled and discrete $E_g$, in

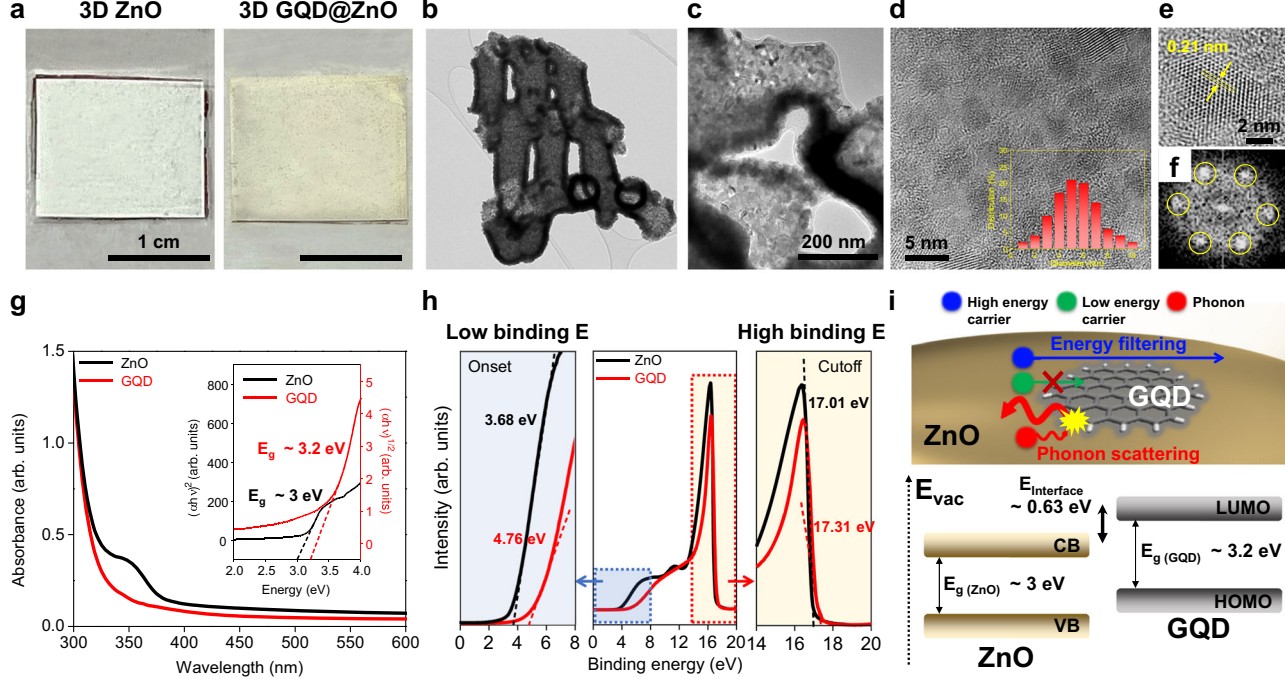

**Fig. 4 | Characterization of 3D GQD@ZnO. a** Digital image of bare 3D ZnO (before decorating with GQDs) and 3D GQD@ZnO (after decorating with GQDs via the solution process). **b** TEM image for a fragment of 3D GQD@ZnO.
**c, d** Magnifications of **b**. The inset in **d** shows the size distribution of GQD.
**e** Magnification of **d** indicating the hexagonal arrangement of GQD with a d-spacing of 0.21 nm. **f** FFT patterns of **e**. Yellow circles are hexagonal FFT patterns of GQD.

**g** UV−Vis absorption spectra of ZnO film and GQDs. The inset shows their Tauc plots. **h** Low and high-binding energies of ZnO film and GQDs, respectively, obtained from ultraviolet photoelectron spectroscopy (UPS) spectra. **i** Schematics of energy filtering and phonon scattering at GQD/ZnO interface based on the calculated energy-band structure.

contrast to highly oxidized GQDs with uncontrollable energy states, enabled precise control over the transport of phonons and electrons through a uniform band alignment between GQD and ZnO.

## Impact of GQD on electron/phonon transport characteristic of 3D ZnO

The afore-described controlled energy barrier at numerous heterogeneous interfaces in 3D GQD@ZnO is expected to significantly enhance the thermoelectric performance, effectively modifying the electron/phonon transport behavior. The 3D ZnO was treated in GQDs solutions with various concentrations. Note that $x = 1, 2, 3$, and 4 at the 3D GQD$x$@ZnO, corresponding to 0.025, 0.05, 0.1, and 0.2 mg mL$^{-1}$ concentrations of GQDs solutions (Supplementary Fig. 10). To verify the impact of the heterogeneous interface on the thermoelectric behavior, we systematically measured the temperature-dependent thermoelectric properties of 3D GQD@ZnO. The $S$ value of 3D GQD$x$@ZnO increased from $-385\,\mu V\,K^{-1}$ (for $x = 0$ for pristine 3D ZnO) to $-556\,\mu V\,K^{-1}$ (for $x = 3$) at 580 K, showing an approximately 44% improvement in $S$ after interfacing GQDs (Fig. 5a). This result indicates effective low-energy electron filtering at numerous heterogeneous interfaces. Note that the value of $S$ depends on the charge carrier concentration ($n$) as $S \propto n^{-2/3}$ [18]. The Hall measurement obtained at 300 K confirmed that $n$ gradually decreased in 3D GQD$x$@ZnO with an increase in the GQD concentration from $x = 0$ to 3 (Supplementary Fig. 11a). The reduction in $n$ with an increase in the GQD content was ascribed to the formation of an interfacial energy barrier of 0.63 eV at

the GQD/ZnO heterogeneous interface, which acted as an energy trap for the conduction electrons within the 3D ZnO [48]. Notably, the improvement in $S$ through the energy filtering effect was generally observed when the potential energy barrier exceeded 0.3 eV, providing further validation of our results [31]. As the temperature increased, the gradual increase in $n$ was observed in 3D GQD@ZnO (Supplementary Fig. 11b). This trend could be attributed to low-energy carriers, which are trapped at the interfacial energy barrier around 300 K, gradually overcoming the barrier due to thermal excitation [48]. Meanwhile, charge carrier mobility ($\mu$) at 300 K showed a gradual decrease in 3D GQD@ZnO with an increase in the GQD concentration (Supplementary Fig. 11a), which is attributed to enhanced impurity scattering. As the temperature increased, $\mu$ gradually decreased in all 3D GQD@ZnO systems due to the enhanced electron-electron collisions with a gradual increase in $n$ (Supplementary Fig. 11c).

Meanwhile, the $\sigma$ value of the 3D GQD$x$@ZnO gradually decreased from 2531 S m$^{-1}$ (for $x = 0$) to 2123 S m$^{-1}$ (for $x = 3$) at 580 K (Supplementary Fig. 11d), indicating charge carrier scattering at numerous heterogeneous interfaces (Supplementary Fig. 11a). However, the increase in $S$ (44%) in the interfacing GQDs (from $x = 0$ to 3) was significantly greater than the reduction in $\sigma$ (16%), indicating that high-energy electrons contribute more to the overall electrical conduction than low-energy electrons. The resultant power factor of the 3D GQD$x$@ZnO system gradually increased from 0.377 mW m$^{-1}$ K$^{-2}$ (for $x = 0$) to 0.657 mW m$^{-1}$ K$^{-2}$ (for $x = 3$) at 580 K, representing a 74% performance improvement in interfacing GQDs (Supplementary

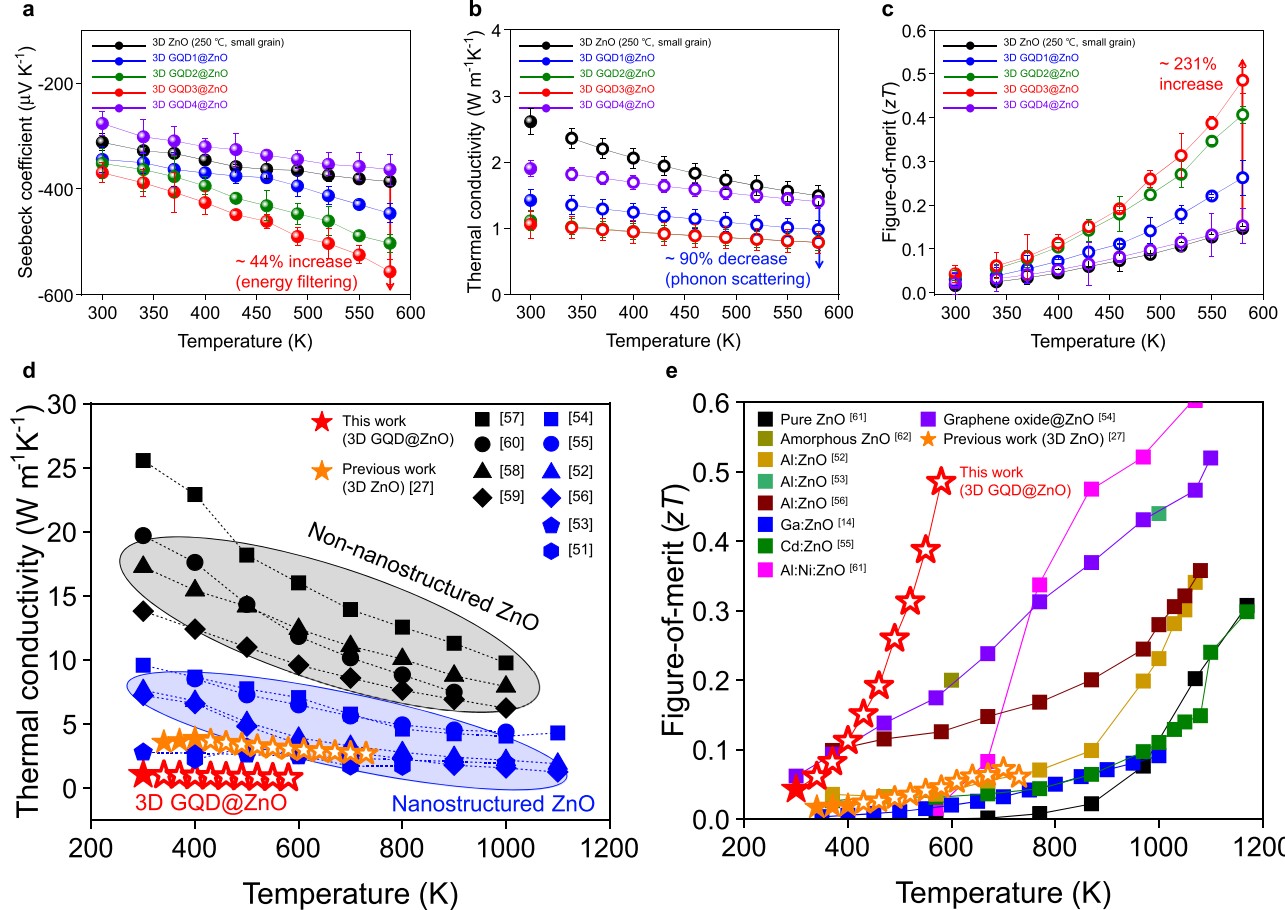

**Fig. 5 | Impact of GQDs on thermoelectric performance of 3D ZnO.** Temperature-dependent **a** Seebeck coefficient, **b** thermal conductivity, and **c** $zT$ of 3D ZnO (small grain) and 3D GQD$x$@ZnO ($x = 1, 2, 3$, and 4). Filled circles are measured values and open circles are estimated values. **d** Comparison of previously reported non-nanostructuring and nanostructuring approaches to the thermal

conductivity of ZnO as a function of temperature. Filled star is measured values and open stars are estimated values. **e** Temperature-dependent $zT$ values of previously reported nanostructured ZnO-based thermoelectric materials. Error bars shown in the Fig. 5 represent the SD.

Fig. 11e). To quantitatively demonstrate the beneficial effect of interfacing GQDs in 3D ZnO on the electronic transport properties, we calculated the weighted mobility for 3D GQD@ZnO (Supplementary Fig. 12). Weighted mobility is a direct indicator of the "electronic" performance of thermoelectric materials. Note that the weighted mobility is insensitive to the carrier concentration of a materials[49], enabling a straightforward assessment of the GQD interfacing effect on the "electronic" performance. We observed a substantial enhancement in the weighted mobility of ZnO upon GQD interfacing, indicating that GQDs effectively improve the electronic transport properties of 3D ZnO through an energy filtering effect. It should be noted that we evaluated the thermoelectric parameters within the temperature range of 300–580 K, considering the thermal stability of GQDs. The temperature-dependent weight loss in 3D GQD@ZnO, as evaluated by thermal gravimetric analysis (TGA), was negligible due to the weight difference between the 3D ZnO and interfaced GQDs. However, the TGA results clearly showed the thermal decomposition of GQDs around 600 K (Supplementary Fig. 13a). Consequently, temperatures beyond 600 K exhibit an abrupt decrease in $S$ owing to the thermal decomposition of GQDs (Supplementary Fig. 13b).

To further investigate the impact of a heterogeneous interface with an interfacial energy barrier on thermal transport, we evaluated the in-plane $\kappa$ of the 3D GQD@ZnO as a function of temperature. For the estimation of $\kappa$ at high temperatures based on the measured value at 300 K, we considered the Umklapp process, normal process, boundary scattering (grain boundary of 3D ZnO), and nanoprecipitation scattering (GQD/ZnO heterogeneous interface). Figure 5b depicts the measured $\kappa$ of the 3D GQD@ZnO (filled circles) at 300 K and the estimated values (open circles) at the 340–580 K temperature range. An abrupt decrease of 90% in $\kappa$ was observed from 1.49 to 0.785 W m$^{-1}$ K$^{-1}$ with an increase in the GQD content from $x = 0$ to 3. The decrease in $\kappa$ resulted from the scattering of phonons with relatively short MFP at numerous heterogeneous interfaces. The $zT$ values of the 3D GQD@ZnO samples were calculated based on a set of measurements as a function of temperature (Fig. 5c). Based on the enhanced power factor and decreased $\kappa$ resulting from structural and interface engineering techniques, the 3D GQD$x$@ZnO ($x = 3$) exhibited the maximum $zT$ of 0.486 at 580 K with high reproducibility of the thermoelectric performance, which was determined by conducting repeated measurements of its thermoelectric properties (Supplementary Fig. 14). Please note that the estimated $zT$ value for 3D GQD$x$@ZnO ($x = 3$) exhibits a potential variation from 0.434 to 0.560 at 580 K, corresponding to a variation in $\kappa$ from 0.879 to 0.681 W m$^{-1}$ K$^{-1}$ (Supplementary Fig. 15), considering the grain size distribution of ZnO within the range of 10 to 60 nm (Fig. 2g). It is noteworthy that the typical measurement error of $zT$ is approximately 20%[50].

The advancements in nanotechnology have significantly improved the thermoelectric performance of materials by effectively decoupling the interconnected nature of individual thermoelectric parameters. This has been achieved through the precise engineering of phonons and electrons at the nanoscale level. For example, we compared the $\kappa$ values of nanostructured ZnO and non-nanostructured ZnO based on literature (Fig. 5d)[27,51–60]. In comparison to non-nanostructured ZnO with micro-level feature sizes, nanostructured ZnO exhibited a significant reduction in $\kappa$ owing to the effective phonon scattering mechanisms. Interestingly, the achieved $\kappa$ value of 3D GQD@ZnO is significantly lower than that of nanostructured ZnO counterparts, suggesting effective phonon scattering at numerous nanoscale interfaces such as 3D nanostructured surfaces, grain boundaries, and heterogeneous interfaces within the 3D GQD@ZnO. With the significant reduction in $\kappa$, improved power factor attributed to the effective low-energy electrons filtering at numerous heterogeneous interfaces results in the highest $zT$ record at 580 K for nanostructured ZnO-based thermoelectric materials to the best of our knowledge (Fig. 5e)[14,27,52–56,61,62]. It should be noted that we intentionally showed the limited $zT$ value of 3D GQD@ZnO below 600 K to ensure the

reliability of the thermoelectric performance of this system, considering the thermal stability of GQDs (Supplementary Fig. 13a, b).

Previous studies suggest that the overloading of nanocarbon materials in bulk samples deteriorates the thermoelectric performance through aggregation and/or a continuous percolation network of nanocarbon materials[32]. In this study, interfacing GQDs with high concentrations of 3D ZnO, 3D GQD$x$@ZnO ($x = 4$), had a detrimental effect on the thermoelectric performance. For the 3D GQD$x$@ZnO ($x = 4$), the measured thermoelectric properties were off the trend when compared with that of $x = 1, 2,$ and 3 (Fig. 5a–c and Supplementary Fig. 11d, e). The thermoelectric properties of the 3D GQD$x$@ZnO ($x = 1, 2, 3,$ and 4) are summarized in Supplementary Table 3. Owing to the strong van der Waals interactions of graphitic nanomaterials, GQDs tend to aggregate easily and/or form a continuous percolation network when used in high concentrations in 3D ZnO. The abrupt increase of $n$ in the 3D GQD$x$@ZnO ($x = 4$) (Supplementary Fig. 11a) indicates that a conduction pathway of π-electrons generated from π-π stacking between GQDs is formed, resulting in the increase in $\sigma$ (Supplementary Fig. 11d). $\kappa$ in the 3D GQD$x$@ZnO ($x = 4$) system also showed an opposite trend compared to that in the $x = 1, 2,$ and 3 systems (Fig. 5b), which is attributed to the increased MFP of phonons within the percolation network formed by GQD overloading (Supplementary Fig. 16). Furthermore, the XRD analysis confirmed a strong peak at approximately 25°, which represents graphitic materials in 3D GQD$x$@ZnO ($x = 4$) when compared with those of $x = 1, 2,$ and 3 (Supplementary Fig. 17). Although the incorporation of nanocarbon materials successfully enhanced the thermoelectric performance, overloading of nanocarbon materials must be avoided. Otherwise, the transport pathways for electrons and phonons could be affected, which could severely deteriorate the thermoelectric performance.

## Discussion

We reported a rational design for achieving high thermoelectric performance by interfacing lowly-oxidized GQDs with a 3D nanostructured ZnO. 3D nanostructuring and appropriate grain boundary engineering significantly enhanced phonon scattering at nanostructured surfaces and grain boundaries, resulting in an exceptionally low thermal conductivity of 1.49 W m$^{-1}$ K$^{-1}$ at 580 K, which is close to the theoretical minimum for ZnO (1.22 W m$^{-1}$ K$^{-1}$). In addition to the structural design, interfacing lowly-oxidized GQDs with the 3D ZnO support formed a precisely controlled interfacial energy barrier owing to the energy offset between the LUMO of the GQD and the CBM of ZnO, effectively allowing for low-energy electron filtering and further phonon scattering at the interfacial energy barrier. The proposed 3D GQD@ZnO exhibited an extraordinarily high $zT$ of 0.486 (estimated value) at 580 K owing to an ultralow thermal conductivity (0.785 W m$^{-1}$ K$^{-1}$ at 580 K, estimated value) and an extremely high Seebeck coefficient (−556 μV K$^{-1}$ at 580 K), outperforming all other ZnO-based materials within the temperature range below 600 K. Our experimental results suggest that multiscale nanostructuring techniques, such as lithography-based structural engineering and interface modification with low-dimensional materials, can provide a rational approach to improve the thermoelectric performances of non-traditional thermoelectric materials, paving the way for new classes of thermoelectric systems. Furthermore, we believe that the proposed approach can be extended to wearable and flexible thermoelectric devices, resulting from the biocompatibility of the GQDs and the unique mechanical properties of 3D ZnO systems. Finally, the lithography-based design of thermoelectric materials developed in this study has the potential for use in cooling applications in electronic devices, given its compatibility with current semiconductor processing technologies.

# Methods

## Preparation of periodic 3D nanostructured polymeric template

The detailed fabrication process for the 3D polymeric template is described in our previous works[21,22,26]. A photoresist (PR) (SU-8, Microchem) with a thickness of ~10 μm was spin-coated at 3000 rpm for 30 s on a sapphire substrate, which was treated by air plasma (CUTEMP, Femtoscience) for 2 min at a flow rate of 45 sccm, pressure of 40 mTorr, and power of 60 W. The spin-coated PR layer was slow-baked at 65 °C for 30 min and at 95 °C for 30 min in two steps. Next, the PR-coated substrate was gently placed in contact with a conformal polydimethylsiloxane phase mask containing a square array of holes with a diameter of ~480 nm, depth of ~420 nm, and periodicity of ~600 nm. Subsequently, a collimated laser (neodymium-doped yttrium aluminum garnet laser: 355 nm, 300 MW, Advanced Opto-wave) was irradiated by passing it over the phase mask. After conventional lithographic procedures (post-baking at a 65 °C hot plate for 7 min, developing (SU-8 developer, Microchem), rinsing with deionized water, and drying), a periodic 3D nanostructured polymeric template was prepared.

## Fabrication of 3D thin-shell ZnO

A thin ZnO layer (70 nm) was conformally deposited on the prepared 3D nanostructured polymeric template via atomic layer deposition at 90 °C (Nexusbe Co., Ltd)[27]. Diethyl zinc (DEZ) (UP Chemical) and $H_2O$ were used as the precursor and reactant, respectively. Each ALD cycle consisted of a sequential process involving a precursor dose (DEZ 100 sccm for 1 s), purging (Ar 100 sccm for 15 s), reactant dose ($H_2O$ 100 sccm for 1.5 s), and purging (Ar 100 sccm for 15 s). The deposition rate was approximately 0.21 nm per cycle. Next, the polymeric component of the ZnO-coated 3D nanostructured template was selectively removed by plasma treatment using a remote plasma etcher (STP Compact, Muegge). Finally, the prefabricated amorphous 3D thin-shell ZnO was annealed in a box furnace at 250 °C (small grain sample) and 500 °C (large-grain sample) for 2 h under atmospheric conditions to control its grain size.

## Preparation of GQD solution and fabrication of 3D GQD@ZnO

The 5 nm GQDs were fabricated from GICs via a previously reported method[36]. Graphite (Bay Carbon Inc.) and potassium sodium tartrate tetrahydrate ($KNaC_4H_4O_6·4H_2O$) were mixed and ground together at a ratio of 1:15 in a mortar. The homogeneous powder mixture was heated at 280 °C in a heating mantle for 12 h to form GICs. The GICs were ground into a fine powder in a vial and immersed in DI water under ultrasonication for 3 h to produce GQDs. The size separation of the GQDs was accomplished using vacuum filtration and centrifugation. The sonicated solution was vacuum filtered through a 0.02-μm Anodisc membrane filter. The filtered GQDs (<20 nm in size) were centrifuged using 30 kDa and 10 kDa molecular weight cutoff centrifugal microfilters (Amicon Ultra-15) to obtain GQDs <5 nm in size. Finally, the as-prepared aqueous solution of GQDs was dialyzed in a dialysis membrane tubing (3.5 kDa) for at least 3 d to completely remove any residual salts. Next, dip-coating was used to prepare the 3D GQD@ZnO using the aqueous solution and 3D ZnO. The prefabricated 3D ZnO was immersed into the GQD solution (0.025 (GQD 1), 0.05 (GQD 2), 0.1 (GQD 3), and 0.2 mg mL$^{-1}$ (GQD 4)) and heated at 60 °C for 6 h for sufficient coating of GQDs onto the ZnO surface. Finally, the 3D GQD@ZnO was obtained after drying the immersed 3D ZnO via heat treatment (70 °C, 1 h).

## Characterization

The morphology of the 3D ZnO was characterized using a field-emission scanning electron microscope (FE-SEM, Magellan 400, FEI Co.), operated at 5–10 kV. The nanostructural features of the 3D GQD@ZnO, including submicron pores, grain boundaries, and GQD/ZnO interfaces, were obtained by field-emission transmission electron microscopy (FE-TEM, Titan cubed G2 60-300, FEI Co.) at 200 kV. The morphology and thickness of GQDs were characterized using atomic force microscopy (AFM, INNOVA-LABRAM HR800) in the tapping mode under ambient conditions. Crystallographic analyses of the samples were performed using high-resolution X-ray diffraction (XRD; RIGAKU). The chemical bonds of the samples were investigated using X-ray photoelectron spectroscopy (XPS, K-alpha, Thermo VG Scientific) and Fourier-transform infrared spectroscopy (FT-IR, Nicolet iS50, Thermo Fisher Scientific Instruments). The absorption spectra were obtained using a UV/Vis/NIR spectrophotometer (SolidSpec-3700, Shimadzu) with an integrating sphere. The electronic band structures of the ZnO thin films (VB and CB) and GQDs (HOMO and LUMO) were determined using ultraviolet photoelectron spectroscopy (UPS, Sigma Probe, Thermo VG Scientific) and the Tauc plot method. The quantitative composition of GQDs was estimated by thermal gravimetric analysis (TGA; Mettler Toledo).

## Measurement of thermoelectric properties

The temperature-dependent electrical conductivity ($σ$) and Seebeck coefficient ($S$) of the 3D ZnO and 3D GQD@ZnO were measured using a Linseis (LSR-3) system under a He atmosphere. $σ$ was calculated using the following formula (1):

$$σ = \frac{l}{A \cdot R} \tag{1}$$

Here, $R$, and $A$ are the resistance and cross-sectional area of the sample, respectively, and the distance between the two probes is represented by $l$ (Supplementary Fig. 18). Note that the measured $σ$ values needed to be modified considering the shape correction factor, especially considering the effects of porosity ($\phi$). The measured $σ$ value was refined to $σ_{3D}$ using the following Eq. (2)[63]:

$$σ_{3D} = σ/3 \times [(1-\phi) + 2(1-\phi)^{1.5}] \tag{2}$$

To measure the temperature-dependent $σ$ and $S$ values, the heating rate was set to 2 °C min$^{-1}$ with a 5 °C temperature gradient across a 3-mm-long sample. The temperature-dependent $σ$ and $S$ values for all samples were measured five times with different samples to ensure their reliability. The temperature-dependent carrier concentration and mobility were measured using a high-temperature Hall effect measurement system (HMS-8407, Lake Shore Cryotronics Inc.).

## Measurement of thermal conductivity

The suspended-type TE measurement platform (MTMP) consisted of a microheater to generate a temperature gradient and a pair of Pt thermometers to read the temperature difference in the sample. The thermal conductance ($G$) of 3D ZnO and 3D GQD@ZnO were measured under vacuum conditions ($10^{-6}$ Torr) at 300 K. $G$ values obtained via the MTMP system were converted into thermal conductivity ($κ$) considering the cross-sectional area ($A$) and length ($l$) of the samples using the following Eq. (3):

$$κ = \frac{G \cdot l}{A} \Delta T \tag{3}$$

$κ$ for all samples was measured five times with different samples at 300 K to ensure its reliability.

## Estimation of lattice thermal conductivity

The lattice thermal conductivity at room temperature was calculated by subtracting the electronic contribution of thermal conductivity ($κ_e$)

from total thermal conductivity ($\kappa_t$) using the following Eq. (4):

$$\kappa_e = L\sigma T \tag{4}$$

where $L$, $\sigma$, and $T$ are the Lorenz number, electrical conductivity, and absolute temperature, respectively. The Lorenz number was calculated from the experimentally measured Seebeck coefficient ($S$) using the following Eq. (5)[64]:

$$L = 1.5 + \exp\left[-\frac{|S|}{116}\right] \tag{5}$$

where $L$ is in $10^{-8}$ W$\Omega$K$^{-2}$ and $S$ in $\mu$V K$^{-1}$. This equation provides a satisfactory approximation (uncertainty within 5%) when electron transport is dominated by acoustic phonon scattering. The temperature-dependent lattice thermal conductivity was estimated using a modified Debye-Callaway model using the following Eq. (6):

$$\kappa_{latt} = \frac{\kappa_B}{2\pi^2 v}\left(\frac{\kappa_B T}{\hbar}\right)^3 \int_0^{\theta_{D/T}} \tau_{tot}(x)\frac{x^4 e^x}{(e^x - 1)} dx \tag{6}$$

where $k_B$, $v$, $\hbar$, $\theta_D$, and $\tau_{tot}$ are the Boltzmann constant, average sound velocity, Planck constant, Debye temperature, and combined relaxation time of the phonon, respectively. We considered the Umklapp, normal, boundary scattering, and nanoprecipitation scattering processes. The relaxation time for each process is given as following Eqs. (7–9):

$$\tau_U^{-1} + \tau_N^{-1} = \beta\frac{2}{(6\pi^2)^{1/3}}\frac{k_B \bar{V}^{1/3}\gamma^2\omega^2 T}{\bar{M}v^3} \tag{7}$$

(Umklapp and normal process)

$$\tau_B^{-1} = \frac{v}{d} \tag{8}$$

(Boundary scattering)

$$\tau_I^{-1} = v_s N_p\left[\left((2\pi R^2)^{-1} + \left(\frac{4}{9}\pi R^2\left(\frac{\Delta D}{D}\right)^2\left(\frac{\omega R}{v_s}\right)^4\right)^{-1}\right)\right]^{-1} \tag{9}$$

(Nanoprecipitate scattering)

The total relaxation time is given by Matthiessen's rule as following Eq. (10):

$$\tau_{tot}^{-1}(x) = \tau_U^{-1}(x) + \tau_N^{-1}(x) + \tau_B^{-1} + \tau_I^{-1} \tag{10}$$

Finally, the effect of porosity was considered based on the effective medium theory as follows Eq. (11):

$$\kappa_{l,porous} = (1 - P)^{3/2}\kappa_{l,dense} \tag{11}$$

where $P$ is the porosity of the given material. The material parameters used to estimate the high-temperature thermal conductivities of the 3D ZnO and 3D GQD@ZnO are listed in Supplementary Table 4.

## Reporting summary
Further information on research design is available in the Nature Portfolio Reporting Summary linked to this article.

## Data availability
The authors declare that all data supporting the findings of this study are available within the article and its Supplementary Information file.

Further datasets are available from the corresponding author upon reasonable request.

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

## Acknowledgements

S.J. appreciates the financial support by the National Research Foundation of Korea (NRF) grants (NRF-2020M3D1A1110522, NRF-2022M3H4A1A04068923, and RS-2023-00256286) funded by the

Ministry of Science and ICT (MSIT) of the Korean governments. S.J. also appreciates the support from the Characterization Platform for Advanced Materials (KRISS-2022-GP2022-0013) funded by the Korea Research Institute of Standards and Science.

## Author contributions

M.C. and J.A. contributed equally to this study. M.C. and S.J. conceived and designed experiments. M.C. and J.A. prepared the 3D thin-shell ZnO, GQDs solution, and 3D GQD@ZnO. M.C., J.A. and D.C. performed the SEM, AFM, XRD, XPS, UPS, UV–Vis and TGA for material characterization. J.H.P. and S.M.K. performed HR-TEM for the crystallinity analysis of samples. M.C. measured the electrical properties of samples. H.L., J.Y.S. and H.S. measured the thermal conductivities of samples. H.J. and M.–W.O. calculated the high-temperature thermal conductivities. All authors provided insights into the data analysis. M.C. and J.A. wrote the manuscript. H.S. and S.J. supervised the study. All the authors approved the final version of the manuscript.

## Competing interests

The authors declare no competing interests.
