## [Peer Review File · Nature Communications]

REVIEWER COMMENTS

Reviewer #1 (Remarks to the Author):

The article titled "High Figure-of-merit for ZnO Nanostructure by Interfacing Lowly-Oxidized Graphene Quantum Dots" presents a highly innovative approach addressing a crucial aspect related to enhancing the figure of merit and, consequently, the efficiency of thermoelectric systems. Nanoengineering is underscored as a potent solution, as elucidated in this study. The strategy of integrating ZnO nanostructures with the interface of lowly-oxidized graphene quantum dots proved to be notably effective. In my assessment, this outcome is groundbreaking and holds potential for publication, albeit with potential for substantial revisions.

In addition, numerically delineating the following concerns:

1. In the introduction, the authors refrain from employing suitable references to compare with other materials operating in the same operational zone. Incorporating relevant literature in this section could enhance the contextualization.
2. Further exploration is warranted regarding the reduction of thermal conductivity. For instance, a more comprehensive examination, such as comparing the theoretical phonon mean free path for ZnO with its actual dimensions, could provide greater clarity.
3. The methodology for determining electrical conductivity requires elucidation. Specifically, details on the shape correction factor should be expounded upon, as discerning how to correct this factor in the present case proves challenging. Is the conductivity following a classical regime, or is there already a quantum contribution?
4. A notable issue with this article is that the authors have extensively published on this technique. This work appears to be an incremental step, leveraging the same methodology for a different application. Clarification on how this work represents a significant advancement or departure from their prior publications is necessary.
- 5 The methodology for determining error bars warrants clarification. Are they a result of instrument resolution, fluctuations, or the repetition of measurements? Elaborating on the precise procedures employed in determining error bars will enhance the transparency and robustness of the reported results.
- 6 Finally, it would be beneficial for the authors to provide more practical insights into potential applications and their vision for implementing this technology in a practical device. Expanding on the

practical implications and envisaged implementation of the findings could significantly enhance the article's impact. Otherwise, it risks being perceived as yet another materials-focused article with a focus on the thermoelectric figure of merit (ZT).

Reviewer #2 (Remarks to the Author):

Brief summary:

The authors have developed a nanostructured zinc oxide material as a thermoelectric material with low environmental load. Three steps were used as a strategy to improve the performance. First, the material itself is processed into a porous three-dimensional nano-periodic structure by using novel patterning technique to effectively scatter phonons with long MFPs and achieve lower thermal conductivity than the bulk ZnO. Next, thermal annealing was used to control the ZnO grain size, which was optimized to improve the electrical properties. Finally, nanoscale flack-shape graphene was modified on the surface to achieve high thermoelectric figure of merit ZT by reducing thermal conductivity due to scattering of phonons with short MFPs and increasing Seebeck coefficient by filtering low energy electrons.

It reports the lowest thermal conductivity and highest figure of merit ZT compared to other nanostructured ZnO thermoelectric materials.

It is worth mentioning that a very precise structural and composition analysis of the prepared samples was performed. The temperature dependent measurement of Seebeck coefficient and electrical conductivity, as well as the measurement of thermal conductivity and the estimation of its temperature dependence by theoretical calculation models, are also carefully performed and reasonable.

Overall, I believe that publication could be considered by revising some of the expressions or, if possible, conducting additional experiments.

Comments:

If high performance at high temperatures is to be strongly claimed, I believe that experimental measurement of thermal conductivity is essential. ($ZT = 0.486$ and $\kappa = 0.785 \text{ Wm}^{-1}\text{K}^{-1}$ at 580 K)

If these values are claimed in abstract and conclusion, they should be mentioned as estimated values.

If additional experiments are difficult, one option would be to focus on grain size control by annealing and the concentration dependence of graphene quantum dots.

I think it is unfair to discuss only at 580 K temperature in comparison with other materials. While the structure authors reported can break down at temperatures above 600 K, it is hard to consider using the material in exactly 580 K hot environment for actual applications.

Or do you have any idea to improve temperature stability of GNDs@3D ZnO above 600 K?

In page 8, regarding the three-dimensional structure, the authors state that they have optimized the dimensions, but only one dimension (~ 70 nm) is reported in the reference 24 (10.1039/c7nr08167g) and the optimization needs to be explained.

I believe that thermoelectric properties are strongly dependent on dimensions in this scale, especially thermal conductivity.

About the model for temperature dependence of thermal conductivity, rate for boundary scattering is defined as v/d , and 20 nm is used for d according to the supplemental information. Because this κ value is the basis for the main claim, I think it is inaccurate to use only one value. At least, it should have some range with in range of $d = 10-60$ nm according to the grain distribution for ZnO annealed at 250 degree.

Reviewer #3 (Remarks to the Author):

This manuscript reports a significantly high zT of 0.486 at 580 K in the 3D nanostructured ZnO with graphene quantum dots interfaced. The 3D nanostructured ZnO was fabricated by atomic-layer depositing the thin ZnO layer on the 3D nanostructured polymeric template. The structural design is very interesting and leads to a substantial reduction in κ without degrading σ , but it seems to have been reported many times (such as the references 24-26). In addition to the structural engineering, interfacing lowly-oxidized GQDs with the 3D ZnO support effectively allows for low-energy electron filtering and further phonon scattering at the interfacial energy barrier, which enhances S , and also leads to the significant reduction in κ from 1.49 to 0.785 W m⁻¹K⁻¹. Some concerns are listed below:

1. Please show the repeatability of high thermoelectric properties of the 3D GQDs@ZnO 3 sample.
2. Discussion on the temperature-dependent carrier concentration and Hall mobility is needed.
3. Please show the thermal gravimetric analysis result for the sample of 3D GQDs@ZnO 3 within the temperature range of 300–580 K.
4. Since the highest zT achieved in this work outperforms any other ZnO-based materials reported to date, it is necessary to show the κ measurement result at high temperatures rather than estimated data.

<Responses to the reviewers' comments>

Manuscript ID: NCOMMS-23-41901

Title: High Figure-of-merit for ZnO Nanostructures by Interfacing Lowly-Oxidized Graphene Quantum Dots

We appreciate the valuable comments from the reviewers. We have revised the manuscript according to the reviewer's directions. Thank you very much.

REVIEWER REPORT(S):

Reviewer #1:

General Comments: *The article titled "High Figure-of-merit for ZnO Nanostructure by Interfacing Lowly-Oxidized Graphene Quantum Dots" presents a highly innovative approach addressing a crucial aspect related to enhancing the figure of merit and, consequently, the efficiency of thermoelectric systems. Nanoengineering is underscored as a potent solution, as elucidated in this study. The strategy of integrating ZnO nanostructures with the interface of lowly-oxidized graphene quantum dots proved to be notably effective. In my assessment, this outcome is groundbreaking and holds potential for publication, albeit with potential for substantial revisions. In addition, numerically delineating the following concerns:*

Our response to General Comments: We appreciate the reviewer's assessment of the novelty of this work and the overall positive comments. Our responses to the reviewer's concerns are provided below. We hope that these revisions make our manuscript more understandable and informative for the readers. Thank you again.

Topic 1: *In the introduction, the authors refrain from employing suitable references to compare with other materials operating in the same operational zone. Incorporating relevant literature in this section could enhance the contextualization.*

Our response to topic 1: Thank you for your valuable comment. Zinc oxide (ZnO) is a representative thermoelectric metal oxide with a wide operating temperature (570–1500 K). Conventional thermoelectric materials, including PbTe-based alloys (operating temperature range of 600–900 K) and SiGe-based alloys (operating temperature above 1000 K), operate within temperature ranges similar to that of ZnO (*Ceram. Soc. Jpn.*, 2011, **119**, 770-775). While these conventional thermoelectric materials exhibit outstanding thermoelectric performances ($zT > 1.5$), their practical use is limited because of their toxicity and scarcity. In this regard, investigating thermoelectric metal oxides is crucial, given their advantages including nontoxicity, earth-abundance, and high thermal and chemical stability, as compared to conventional thermoelectric materials.

Typically, thermoelectric metal oxides reach their maximum zT (zT_{\max}) at high temperatures (nearly 1000 K). Similarly, ZnO exhibits its zT_{\max} at high temperatures. However, we intentionally reported the limited zT of 3D GQD@ZnO below 600 K considering the thermal stability of GQDs, as GQDs begin to thermally decompose at 600 K (**Fig. R1**). Following the reviewer's suggestions, we have explored the thermoelectric properties of various thermoelectric metal oxides (e.g., SrTiO₃, In₂O₃, Ca₃Co₄O₉, CaMnO₃, and Na_xCo₂O₄), which have operating ranges similar to that of ZnO (Table R1). Next, we examined their performances at both the optimal temperature ($T_{zT_{\max}}$) and a mid-temperature (580 K) to compare them with our 3D GQD@ZnO system. As expected, all metal oxides exhibited their zT_{\max} values at a high temperature (~1000 K). Notably, the zT_{\max} values of most oxides were lower than that of 3D GQD@ZnO ($zT_{\max} = \sim 0.486$ at 580 K). This implies the potential for 3D GQD@ZnO to achieve even higher zT values if the GQDs remain stable at higher temperatures. Therefore, our follow-up study will focus on enhancing the thermal stability of 3D GQD@ZnO at even higher temperatures through GQD encapsulation using an additional protective layer. The relevant discussions have been added to the revised manuscript.

Fig. R1| Thermal gravimetric analysis (TGA) results of GQDs.

Types	Materials	T (K)	Figure-of-merit (zT)	Power factor ($S^2\sigma$) ($\mu W cm^{-1} K^{-2}$)	Thermal conductivity (k) ($W m^{-1} K^{-1}$)	Ref.	
SrTiO ₃	Sr _{0.93} La _{0.07} Ti _{0.93} Nb _{0.07} O ₃	T _{zTmax}	1009	0.43	12.6	~3.0	[1]
		T _{mid}	580	~0.29	~18	~3.5	
	SrTi _{0.85} Nb _{0.15} O ₃	T _{zTmax}	1100	0.4	11	3.2	[2]
		T _{mid}	580	~0.14	~10	~4.25	
Sr _{0.9} La _{0.067} TiO ₃ /0.6 wt% graphene	T _{zTmax}	1023	0.36	7.5	2.1	[3]	
	T _{mid}	580	~0.21	~15	~4.1		
In ₂ O ₃	In _{1.88} V _{0.12} O ₃	T _{zTmax}	973	0.42	7.81	~1.5	[4]
		T _{mid}	580	~0.09	~3.5	~2.25	
	In _{1.98} Co _{0.02} O ₃	T _{zTmax}	1073	0.26	4.6	1.8	[5]
		T _{mid}	580	~0.04	~1.5	~3	
In _{1.99} Ge _{0.01} O ₃	T _{zTmax}	1000	0.18	-	3.6	[6]	
	T _{mid}	580	~0.05	-	~6.3		
Ca ₃ Co ₄ O ₉	Sr-doped Ca ₃ Co ₄ O ₉	T _{zTmax}	1073	0.29	11.6	4.4	[7]
		T _{mid}	580	~0.06	~4.5	-	
	Ca ₃ Co ₄ O ₉ from Co ^{II} /Co ^{III} -LDH	T _{zTmax}	1000	0.18	3.1	1.6	[8]
		T _{mid}	580	-	-	-	
CaMnO ₃	Ca _{0.9} Dy _{0.1} MnO ₃	T _{zTmax}	1000	0.2	3.56	~1.85	[9]
		T _{mid}	580	~0.1	~3.52	-	
	Ca _{0.9} Yb _{0.1} MnO ₃	T _{zTmax}	1000	0.2	~3	1.6	[9]
		T _{mid}	580	~0.08	~2.63	-	
Na _x Co ₂ O ₄	Na _x Co ₂ O ₄ /Ag composites	T _{zTmax}	~970	~0.5	~1.4	~2.75	[10]
		T _{mid}	580	~0.2	~1.0	~2.7	
	Na _x Co ₂ O ₄ /Au composites	T _{zTmax}	~970	~0.44	~1.2	~2.8	[10]
		T _{mid}	580	~0.15	~0.85	~2.8	
ZnO	3D GQD@ZnO	T _{zTmax}	580	~0.486	~6.57	~0.785	This work

Table R1| Thermoelectric properties of various metal oxides reported in literature. [1] *J. Mater. Chem. C*, 2018, **6, 7594-7603. [2] *J. Mater. Chem. C*, 2015, **3**, 11406-11411. [3] *ACS Appl. Mater. Interfaces*, 2015, **7**, 15898-15908. [4] *ACS Appl. Energy Mater.*, 2019, **3**, 1552-1558. [5]. *Am. Ceram. Soc.*, 2010, **93**, 2938-2941. [6] *J. Eur. Ceram. Soc.*, 2015, **35**, 145-151. [7] *J. Eur. Ceram. Soc.*, 2019, **39**, 1186-1192. [8] *Mater. Res. Bull.*, 2012, **47**, 3287-3291. [9] *J. Appl. Phys.*, 2008, **104**, 093703 [10] *J. Alloys Compd.*, 2008, **450**, 494-498.**

Modification to the manuscript:

Page 3, added and modified sentences:

Over the past decades, various thermoelectric materials including metal tellurides (e.g., Bi₂Te₃,⁵ PbTe,⁶ etc), metal silicides (e.g., Mg₄Si₇,⁷ Mg₂Si-Mg₂Sn,⁸ etc), and Si-Ge alloys⁹ with high zT values (>1.5) have been actively reported. Although these materials bring thermoelectric technology closer to commercialization, their toxicity, scarcity, and chemical instability remain open challenges. In this regard, metal oxides are potential alternatives because of their nontoxicity, earth-abundance, and high thermal stability, thereby opening up more possibilities for the commercialization of thermoelectric technology.¹⁰ Various metal oxides, such as SrTiO₃, Ca₃Co₄O₉, and Na_xCo₂O₄, have been developed to achieve competitive zT values with conventional thermoelectric materials, as summarized in **Supplementary Table 1**. However, the synthesis techniques for metal oxides with high thermoelectric performances are quite challenging due to their complex chemical configurations and crystal structures.

Page 3, added and modified sentences:

Zinc oxide (ZnO), a well-known n-type thermoelectric metal oxide, has emerged as a promising candidate due to its relatively high S (-100 to -400 $\mu\text{V K}^{-1}$), wide working temperature (570–1500 K), and facile processability compared to other metal oxides.^{11,12}

Page 24, added references:

7. Li Z, Dong J-F, Sun F-H, Hirono S, Li J-F. Significant enhancement of the thermoelectric performance of higher manganese silicide by incorporating MnTe nanophase derived from Te nanowire. *Chem. Mater.* **29**, 7378-7389 (2017).
8. Liu W, *et al.* Convergence of conduction bands as a means of enhancing thermoelectric performance of n-type Mg₂Si_{1-x}Sn_x solid solutions. *Phys. Rev. Lett.* **108**, 166601 (2012).
9. Yu B, *et al.* Enhancement of thermoelectric properties by modulation-doping in silicon germanium alloy nanocomposites. *Nano Lett.* **12**, 2077-2082 (2012).

Page S-20, added a supplementary table:

Types	Materials	Figure-of-merit (zT)	T (K)	Power factor ($S^2\sigma$) ($\mu\text{W cm}^{-1} \text{K}^{-2}$)	Thermal conductivity (κ) ($\text{W m}^{-1} \text{K}^{-1}$)	Ref.
SrTiO ₃	Sr _{0.93} La _{0.07} Ti _{0.93} Nb _{0.07} O ₃	0.43	1009	12.6	~3.0	1
	SrTi _{0.85} Nb _{0.15} O ₃	0.4	1100	11	3.2	2
	Sr _{0.9} La _{0.067} TiO ₂ /0.6 wt% graphene	0.36	1023	7.5	2.1	3
In ₂ O ₃	In _{1.88} V _{0.12} O ₃	0.42	973	7.81	~1.5	4
	In _{1.98} Co _{0.02} O ₃	0.26	1073	4.6	1.8	5
	In _{1.99} Ge _{0.01} O ₃	0.18	1000	-	3.6	6
Ca ₃ Co ₄ O ₉	Sr-doped Ca ₃ Co ₄ O ₉	0.29	1073	11.6	4.4	7
	Ca ₃ Co ₂ O ₉ from Co ^{II} /Co ^{III} -LDH	0.18	1000	3.1	1.6	8
CaMnO ₃	Ca _{0.9} Dy _{0.1} MnO ₃	0.2	1000	3.56	~1.85	9
	Ca _{0.9} Yb _{0.1} MnO ₃	0.2	1000	~3	1.6	9
Na _x Co ₂ O ₄	Na _x Co ₂ O ₄ /Ag composites	~0.5	~970	~1.4	~2.75	10
	Na _x Co ₂ O ₄ /Au composites	~0.44	~970	~1.2	~2.8	10

Supplementary Table 1| Thermoelectric properties of various metal oxides in previous studies.

Page S-24, added supplementary references:

1. Li, Jian-Bo, *et al.* Broadening the temperature range for high thermoelectric performance of bulk polycrystalline strontium titanate by controlling the electronic transport properties. *J. Mater. Chem. C*, **6**, 7594-7603 (2018)
2. Zhang, Boyu, *et al.* High thermoelectric performance of Nb-doped SrTiO₃ bulk materials with different doping levels. *J. Mater. Chem. C*, **3**, 11406-11411 (2015)
3. Lin, Yue, *et al.* Thermoelectric power generation from lanthanum strontium titanium oxide at room temperature through the addition of graphene. *ACS Appl. Mater. Interfaces*, **7**, 15898-15908 (2015)
4. Ahmad, Abid, *et al.* Thermoelectric performance enhancement of vanadium doped n-type In₂O₃ ceramics via carrier engineering and phonon suppression. *ACS Appl. Energy Mater.*, **3**, 1552-1558 (2019)
5. Liu, Yong, *et al.* Effect of transition-metal cobalt doping on the thermoelectric performance of In₂O₃ ceramics. *Am. Ceram. Soc.*, **93**, 2938-2941 (2010)
6. Combe, Emmanuel, *et al.* Microwave sintering of Ge-doped In₂O₃ thermoelectric ceramics prepared by slip casting process. *J. Eur. Ceram. Soc.*, **35**, 145-151 (2015)
7. Torres, M. A., *et al.* Significant enhancement of the thermoelectric performance in Ca₃Co₄O₉ thermoelectric materials through combined strontium substitution and hot-pressing process. *J. Eur. Ceram. Soc.*, **39**, 1186-1192 (2019)

8. Delorme, Fabian, *et al.* Synthesis of thermoelectric $\text{Ca}_3\text{Co}_4\text{O}_9$ ceramics with high ZT values from a $\text{Co}^{\text{II}}\text{Co}^{\text{III}}$ -Layered Double Hydroxide precursor. *Mater. Res. Bull.* **47**, 3287-3291 (2012)
9. Wang, Yang, Yu Sui, and Wenhui Su. High temperature thermoelectric characteristics of $\text{Ca}_{0.9}\text{R}_{0.1}\text{MnO}_3$ (R= La, Pr, ..., Yb). *J. Appl. Phys.* **104** (2008)
10. Ito, Mikio, and Daisuke Furumoto. Effects of noble metal addition on microstructure and thermoelectric properties of $\text{Na}_x\text{Co}_2\text{O}_4$. *J. Alloys Compd.* **450**, 494-498 (2008)

Topic 2: Further exploration is warranted regarding the reduction of thermal conductivity. For instance, a more comprehensive examination, such as comparing the theoretical phonon mean free path for ZnO with its actual dimensions, could provide greater clarity.

Our response to topic 2: Thank you for your constructive comment. As highlighted by the reviewer, we conducted a comparison between the theoretical phonon mean free path (MFP) for ZnO and 3D GQD@ZnO considering the thermal conductivity. We categorized the phonons contributing to the thermal conductivity of ZnO based on three MFP ranges. Phonons with MFPs of 20 nm or less contribute to approximately 8%, those with MFPs between 20 and 100 nm contribute to approximately 36%, and those with MFPs between 100 and 1000 nm contribute to approximately 41% of the thermal conductivity (*J. Mater. Chem. A*, 2020, **8**, 11948-11957). Consequently, phonons with MFPs below 1000 nm are considered to contribute to approximately 85% of the overall thermal conductivity of ZnO (Fig. R2a). In this study, we proposed three strategies to decrease the thermal conductivity of ZnO (Fig. R2b): (1) 3D nanostructuring with pore sizes of several hundreds of nm (resulting in phonon scattering at nanostructured surfaces), (2) grain boundary engineering with grain sizes of several tens of nm (resulting in phonon scattering at grain boundaries), and (3) GQD interfacial scattering with several nm of precipitation (resulting in phonon scattering at nanoprecipitation). These three strategies are expected to effectively scatter phonons with the three categorized phonons, leading to a remarkably low thermal conductivity ($0.785 \text{ W m}^{-1}\text{K}^{-1}$ at 580 K) (Fig. R2c). For 3D GQD@ZnO, while the charge carrier scattering at multiple interfaces led to a slight decrease in electrical conductivity, the remarkably low thermal conductivity resulting from effective phonon scattering contributed to achieving the highest reported thermoelectric performance among ZnO materials to date ($zT \sim 0.486$ at 580 K). The relevant discussions have been added to the revised manuscript.

Fig. R2| a, Cumulative lattice thermal conductivity against the phonon MFP for ZnO (*J. Mater. Chem. A*, 2020, **8**, 11948-11957). **b**, Scattering of phonons with various MFPs at the 3D GQD@ZnO nanostructure. **c**, Thermal conductivity of the bulk ZnO (*ACS Appl. Mater. Interfaces*, 2021, **13**, 23771-23786), 3D ZnO with large grains, 3D ZnO with small grains, and 3D GQD@ZnO.

Modification to the manuscript:

Page 8, added sentences:

To optimize the shell thickness of the 3D ZnO with respect to phonon transport, we prepared three types of 3D ZnO systems with shell thicknesses of 30, 50, and 70 nm by adjusting the deposition cycles of the ALD process. The average pore sizes for three types of 3D ZnO systems were 280, 230, and 180 nm, respectively, which depend on the shell-thickness of ZnO (**Supplementary Fig. 2a, b**). The reduction ratios of the thermal conductivity achieved by phonon scattering in 3D ZnO with different shell thicknesses are expected to be similar, given the comparable contribution of thermal conductivity by phonons with MFPs corresponding to different pore sizes.⁴² However, when the ZnO shell thickness exceeds 100 nm, the thermal conduction achieved by phonons with a short MFP, which are transported within the relatively thick shell (100 nm or greater), becomes more significant than the phonon scattering at the nanostructured surfaces.⁴² Meanwhile, as the shell thickness increases, the 3D ZnO nanostructure becomes more uniform and exhibits high structural stability (**Supplementary Fig. 2c**). For the 30-nm sample with a relatively thin shell, structural stability was relatively lower due to a slight collapse during the removal of the polymer template after ZnO deposition.

Page 10-11, added and modified sentences:

This result suggests enhanced phonon scattering with an increase in the grain boundary fraction, especially for phonons with an MFP in the range of several tens of nm. All the 3D ZnO samples showed a significantly reduced κ when compared with that of the ZnO bulk materials ($> 40 \text{ W m}^{-1}\text{K}^{-1}$), implying the significantly enhanced scattering of phonons, which have an MFP in the range of several hundreds of nm, at numerous 3D nanostructured interfaces.

Page 26, added reference:

42. Spooner KB, Ganose AM, Scanlon DO. Assessing the limitations of transparent conducting oxides as thermoelectrics. *J. Mater. Chem. A* **8**, 11948-11957 (2020).

Supplementary Figure 2 | Optimization of the shell thickness of 3D ZnO in terms of thermal conductivity and structural stability. a, Schematic illustration of 3D thin-shell ZnO. b, Pore sizes of 3D ZnO with 30, 50, and 70 nm shell thickness. c, Structural stability of 3D ZnO nanostructures according to the shell thickness. Error bars shown in the Supplementary Fig. 2 represents the standard deviation (SD).

Topic 3: *The methodology for determining electrical conductivity requires elucidation. Specifically, details on the shape correction factor should be expounded upon, as discerning how to correct this factor in the present case proves challenging. Is the conductivity following a classical regime, or is there already a quantum contribution?*

Our response to topic 3: We apologize for providing insufficient information. The temperature-dependent electrical resistance (R) of the 3D ZnO was measured using a commercial thin-film thermoelectric measurement system (LSR-3, Linseis) under a He atmosphere. The electrical conductivity (σ) was then calculated using the formula ($\sigma = l/A \cdot R$) depending on the cross-sectional area (A) and resistance (R) of the sample and the distance between the two probes (l) (**Fig. R3a**). Compared to film-type materials, our 3D ZnO system exhibits a highly porous network with approximately ~80% porosity (ϕ) (*Adv. Sci.*, 2021, **8**, 2001883 – our previous work). Therefore, the measured σ value needed to be modified considering the shape correction factor, in particular considering ϕ . We refined σ to σ_{3D} using the following equation: $\sigma_{3D} = \sigma/3 \times [(1 - \phi) + 2(1 - \phi)^{1.5}]$ (*J. Porous Mater.*, 2009, **16**, 675-681), which relies on the ϕ value of the 3D ZnO system.

For the 3D GQD@ZnO system, σ was also calculated using the shape correction factor (ϕ) (**Fig. R3b**). The resultant σ slightly decreased in 3D GQD $_x$ @ZnO with an increase in the GQD concentration from $x = 0$ to 3 (at 300 K). This reduction in σ with an increase in the GQD content was ascribed to the formation of an interfacial energy barrier of 0.62 eV at the GQD/ZnO heterogeneous interface, which acted as an energy trap for the conduction electrons within the 3D ZnO. The charge carriers trapped at the GQD/ZnO heterogeneous interface were confirmed by the gradual decrease in charge carrier concentration (n) with an increase in GQD concentration from $x = 0$ to 3 (**Fig. R3c**). In contrast, for the 3D GQD $_x$ @ZnO ($x = 4$) system, n shows a different trend than that of the $x = 1, 2,$ and 3 systems. An abrupt increase of n in the 3D GQD $_x$ @ZnO ($x = 4$) systems indicates that a conduction pathway of π -electrons generated from π - π stacking between GQDs is formed. Meanwhile, the gradual increase in n with increasing temperature (300 to 580 K) was observed in 3D GQD $_x$ @ZnO ($x = 1, 2, 3,$ and 4) (**Fig. R3d**). This result implies that the trapped charge carriers at the interfacial energy barrier around 300 K gradually overcome the energy barrier due to the thermal excitation, exhibiting an increase in σ with temperature. Notably, the release of trapped charge carriers with increasing temperature is generally observed in semiconducting thermoelectric materials (*J. Am. Chem. Soc.*, 2013, **135**, 7486-7495). As a result, considering the charge carriers trap at

the interfacial energy barrier and the gradual increase in n and σ with increasing temperature, it is reasonable to infer that the measured σ involves quantum contributions. The relevant figure and discussions have been added to the revised manuscript.

Fig. R3| a, Measurement system of electrical conductivity in this work. **b**, Temperature-dependent electrical conductivity of 3D ZnO (small grain) and 3D GQD $_x$ @ZnO ($x = 1, 2, 3,$ and 4). Measured charge carrier concentration of the 3D ZnO (small grain) and 3D GQD $_x$ @ZnO ($x = 1, 2, 3,$ and 4) systems at 300 K (**c**) and 300 to 580 K (**d**).

Modification to the manuscript:

Page 14, added and modified sentences:

The Hall measurement obtained at 300 K confirmed that n gradually decreased in 3D GQD $_x$ @ZnO with an increase in the GQD concentration from $x = 0$ to 3 (**Supplementary Fig. 11a**). The reduction in n with an increase in the GQD content was ascribed to the formation of an interfacial energy barrier of 0.62 eV at the GQD/ZnO heterogeneous interface, which acted as an energy trap for the conduction electrons within the 3D ZnO.⁴⁸ Notably, the improvement in S through the energy filtering effect was generally observed when the potential energy barrier exceeded 0.3 eV, providing further validation of our results.³¹ As the temperature increased, the gradual increase in n was observed in 3D GQD@ZnO (**Supplementary Fig. 11b**). This trend could be attributed to low-energy carriers, which are trapped at the interfacial energy barrier around 300 K, gradually overcoming the barrier due to thermal excitation.⁴⁸ Meanwhile, charge carrier mobility (μ) at 300 K showed a gradual decrease in 3D GQD@ZnO with an increase in the GQD concentration (**Supplementary Fig. 11a**), which is attributed to enhanced impurity scattering. As the temperature increased, μ gradually decreased in all 3D GQD@ZnO systems due to the enhanced electron-electron collisions with a gradual increase in n (**Supplementary Fig. 11c**).

Meanwhile, the σ value of the 3D GQD $_x$ @ZnO gradually decreased from 2531 S m⁻¹ (for $x =$

0) to 2123 S m⁻¹ (for $x = 3$) at 580 K (**Supplementary Fig. 11d**), indicating charge carrier scattering at numerous heterogeneous interfaces (**Supplementary Fig. 11a**). However, the increase in S (44%) in the interfacing GQDs (from $x = 0$ to 3) was significantly greater than the reduction in σ (19%), indicating that high-energy electrons contribute more to the overall electrical conduction than low-energy electrons. The resultant power factor of the 3D GQD $_x$ @ZnO system gradually increased from 0.377 mW m⁻¹K⁻² (for $x = 0$) to 0.657 mW m⁻¹K⁻² (for $x = 3$) at 580 K, representing a 74% performance improvement in interfacing GQDs (**Supplementary Fig. 11e**).

Page 20-21, added and modified sentences:

The temperature-dependent electrical conductivity (σ) and Seebeck coefficient (S) of the 3D ZnO and 3D GQD@ZnO were measured using a Linseis (LSR-3) system under a He atmosphere. σ was calculated using following formula:

$$\sigma = \frac{l}{A \cdot R}$$

Here, R , and A are the resistance and cross-sectional area of the sample, respectively, and the distance between the two probes is represented by l (**Supplementary Fig. 18**). Note that the measured σ values needed to be modified considering the shape correction factor, especially considering the effects of porosity (ϕ). The measured σ value was refined to σ_{3D} using the following equation:⁶³

$$\sigma_{3D} = \sigma/3 \times [(1 - \phi) + 2(1 - \phi)^{1.5}]$$

To measure the temperature-dependent σ and S values, the heating rate was set to 2 °C min⁻¹ with a 5 °C temperature gradient across a 3-mm-long sample. The temperature-dependent σ and S values for all samples were measured five times with different samples to ensure their reliability.

Page 27, added reference:

63. Cuevas F, Montes J, Cintas J, Urban P. Electrical conductivity and porosity relationship in metal foams. *J. Porous Mater.* **16**, 675-681 (2009).

Page S-12, added and modified supplementary figure:

Supplementary Figure 11| Hall measurement results of 3D GQD@ZnO. a, Charge carrier concentration and charge carrier mobility of 3D ZnO (small grain) and 3D GQD x @ZnO ($x = 1, 2, 3,$ and 4) at 300 K. **b,** Temperature-dependent charge carrier concentration and **c,** charge carrier mobility of the 3D ZnO (small grain) and 3D GQD x @ZnO ($x = 1, 2, 3,$ and 4). **d,** Temperature-dependent electrical conductivity and **(e)** power factor for 3D ZnO (small grain) and 3D GQD x @ZnO ($x = 1, 2, 3,$ and 4). Error bars shown in the **Supplementary Fig. 11** represents the SD.

Page S-19, added supplementary figure:

Supplementary Figure 18| Measurement system of electrical conductivity for the 3D GQD@ZnO.

Topic 4: *A notable issue with this article is that the authors have extensively published on this technique. This work appears to be an incremental step, leveraging the same methodology for a different application. Clarification on how this work represents a significant advancement or departure from their prior publications is necessary.*

Our response to topic 4: Thank you for your constructive comment. As indicated by the reviewer, we previously demonstrated the fabrication of highly ordered 3D thin-shell ZnO using a proximity field nanopatterning (PnP) technique. In our previous works, the fabricated 3D ZnO nanostructures were utilized as thermoelectric (*Nanoscale*, 2018, **10**, 3046-3052) and piezoelectric (*Nano Energy*, 2020, **78**, 105259) materials. This unique design, particularly in thermoelectric applications, achieved a significant reduction in thermal conductivity (κ) via effective phonon scattering at numerous nanostructured surfaces without degrading electrical conductivity owing to the continuous network, resulting in the enhancement of a figure-of-merit (zT) value (~ 0.072 at 693 K) compared to that of the Bulk ZnO ($\sim 2.9 \times 10^{-3}$ at 693 K) (**Fig. R4a, left** and **Fig. R4b (1)**). However, this zT value is relatively low compared to conventional thermoelectric materials. To overcome this barrier, we newly proposed a multi-step rational strategy for achieving high thermoelectric performance through the grain boundary engineering of 3D ZnO systems and interfacing GQDs on the 3D ZnO structures (3D GQD@ZnO) (Fig. R4a, right). In addition to the structural engineering techniques demonstrated in our previous work, grain boundary engineering in 3D ZnO via careful thermal treatment induced further phonon scattering and low-energy electron filtering at grain boundaries, resulting in a slightly improved zT value at an optimum grain size (~ 30 nm) in thin-shell ZnO (**Fig. R4b (2)**). Furthermore, interfacing lowly-oxidized GQDs with 3D ZnO provides numerous heterogeneous interfaces in GQD/ZnO, forming an interfacial energy barrier. This study indicated that the interfacial energy barrier is the key factor for maximizing the zT value by altering electron and phonon transport. Specifically, the low-energy electrons of ZnO become trapped at the GQD/ZnO interfacial energy barrier, allowing the transport of only high-energy electrons. This interfacial energy barrier also leads to the significant reduction in κ through the nanoprecipitate scattering of phonons. Consequently, the 3D GQD@ZnO exhibits an exceptionally high zT value of 0.486 at 580 K (**Fig. R4b (3)**), yielding a record high value among ZnO-based materials (**Fig. R4c**). This study suggests that the multi-scale nanostructuring strategy could provide a rational approach to improve the thermoelectric performances of non-traditional thermoelectric materials. The relevant discussions have been

added to the revised manuscript.

Fig. R4| a, Novel strategies employed in this study. **b**, Enhancement of thermoelectric performance through (1) 3D nanostructuring, (2) grain boundary engineering of 3D ZnO, and (3) interfacing GQDs in 3D ZnO. **c**, zT values at 580 K of previously reported nanostructured ZnO-based thermoelectric materials.

Modification to the manuscript:

Page 18, added and modified sentences:

Our experimental results suggest that multi-scale nanostructuring techniques, such as lithography-based structural engineering and interface modification with low-dimensional materials, can provide a rational approach to improve the thermoelectric performances of non-traditional thermoelectric materials, paving the way for new classes of thermoelectric systems.

Topic 5: *The methodology for determining error bars warrants clarification. Are they a result of instrument resolution, fluctuations, or the repetition of measurements? Elaborating on the precise procedures employed in determining error bars will enhance the transparency and robustness of the reported results.*

Our response to topic 5: We apologize for providing insufficient information. In this work, the error bars of the thermoelectric properties for grain-size-controlled 3D ZnO and 3D GQD@ZnO were determined via the repeated measurements. We conducted five sets of measurements for the thermoelectric properties of all samples and verified the high thermoelectric performance of 3D GQD_x@ZnO ($x = 3$) with high reproducibility, which is attributed to the high structural uniformity of 3D ZnO and uniform dispersion of GQDs on the interface of ZnO. The relevant figure and discussions have been added to the revised manuscript.

Modification to the manuscript:

Page 15, added and modified sentences:

Based on the enhanced power factor and decreased κ resulting from structural and interface engineering techniques, the 3D GQD_x@ZnO ($x = 3$) exhibited the maximum zT of 0.486 at 580 K with a high reproducibility of the thermoelectric performance, which was determined by conducting repeated measurements of its thermoelectric properties (**Supplementary Fig. 14**).

Page 21, added and modified sentences:

The temperature-dependent σ and S values for all samples were measured five times with different samples to ensure their reliability.

Page 21, added sentence:

The thermal conductance (G) of 3D ZnO and 3D GQD@ZnO were measured under vacuum conditions (10^{-6} Torr) at 300 K. G values obtained via the MTMP system were converted into thermal conductivity (κ) considering the cross-sectional area (A) and length (l) of the samples using the following equation:

$$\kappa = \frac{G \cdot l}{A} \Delta T$$

κ for all samples was measured five times with different samples at 300 K to ensure its reliability.

Page S-15, added supplementary figure:

Supplementary Figure 14| Five sets of measurements for the thermoelectric properties of 3D GQD@ZnO. Temperature dependent (a) Seebeck coefficient, (b) electrical conductivity, (c) power factor, (d) thermal conductivity, and (e) zT values of 3D GQD_x@ZnO (x = 3).

Topic 6: Finally, it would be beneficial for the authors to provide more practical insights into potential applications and their vision for implementing this technology in a practical device. Expanding on the practical implications and envisaged implementation of the findings could significantly enhance the article's impact. Otherwise, it risks being perceived as yet another materials-focused article with a focus on the thermoelectric figure of merit (zT).

Our response to topic 6: We deeply appreciate this valuable comment from the reviewer. We believe that, following a thorough revision, this study can offer valuable insights to researchers in the field of thermoelectric systems by addressing two key aspects: **1) rational design of thermoelectric oxide materials** and **2) potential applications for practical use of these systems**.

1) Rational design of thermoelectric materials

In general, oxides exhibit relatively poor thermoelectric performance compared to conventional thermoelectric materials (e.g., Bi- and Pb-based alloys) due to their inherently low electrical transport properties and high thermal conduction. The interfacing of GQDs in 3D thin-shell ZnO is proposed as an effective approach to overcome the critical bottleneck associated with oxide thermoelectric materials. To quantitatively demonstrate the beneficial effects of interfacing GQDs in 3D ZnO on the electronic transport properties of this system, we calculated the weighted mobility for 3D ZnO and 3D GQD $_x$ @ZnO ($x = 1, 2, 3,$ and 4) (**Fig. R5a**). Weighted mobility is a direct indicator of the “electronic” performance of thermoelectric materials. It is important to note that the weighted mobility is insensitive to the carrier concentration of a material (*Adv. Mater.*, 2020, **32**, 2001537), enabling the straightforward assessment of the GQD interfacing effect on the “electronic” performance. We observed a substantial enhancement in the weighted mobility of ZnO upon the interfacing of GQDs, indicating that GQDs effectively improved the electronic transport properties of 3D ZnO through an energy filtering effect. Notably, the weighted mobility values of 3D GQD $_x$ @ZnO ($x = 3$ and 4) were nearly identical, implying that the optimal condition for 3D GQD $_x$ @ZnO is $x = 3$.

We also evaluated the phonon mean free path (MFP) of 3D ZnO with different grain sizes and 3D GQD $_x$ @ZnO ($x = 1, 2, 3,$ and 4) at 300 K using kinetic theory, where the relationship between the lattice thermal conductivity and phonon MFP is given by the following equation:

$$k_{latt} = \frac{1}{3} C_v v l$$

Here, C_v is the specific heat capacity under constant volume, v is the group velocity of phonons, and l is the phonon MFP. Notably, 3D nanostructuring and grain boundary engineering of ZnO led to a significant reduction in phonon MFP, and further decreases were achieved through GQD interfacing (**Fig. R5b**). Interestingly, as the concentration of GQDs increased, the phonon MFP continued to decrease, except in the case of 3D GQD $_x$ @ZnO ($x = 4$). In this case, the formation of a percolation network of GQDs resulted in a slight increase in phonon MFP, leading to an elevated thermal conductivity and a subsequent degradation in thermoelectric performance.

Our comprehensive calculations suggest that GQD interfacing in 3D ZnO can enhance both the electrical and thermal properties of this system. Furthermore, our strategy may be applied to other oxide thermoelectric materials with intrinsically low electronic transport properties and high thermal conduction. The relevant figure and discussions have been added to the revised manuscript.

Fig. R5| Electron and phonon transport behavior depending on GQD concentration. a, Temperature-dependent weighted mobility of 3D ZnO (small grain) and 3D GQD $_x$ @ZnO ($x = 1, 2, 3,$ and 4). **b,** Phonon mean free path (MFP) for bulk ZnO, 3D ZnO with different grain sizes, and 3D GQD $_x$ @ZnO ($x = 1, 2, 3,$ and 4) at 300 K.

2) Potential applications

We believe that the 3D GQD@ZnO system developed in this study holds the potential for use in flexible energy harvesters based on its unique structural properties. In our previous works, we presented a piezoelectric 3D thin-shell ZnO system, which was fabricated using the same methodology that we employed in the present work and exhibited nearly 3-times the elastic strain limit of bulk ZnO ceramics. Finite element analysis simulations revealed that the 3D thin-shell ZnO recovered by 95.9% upon unloading the compressive pressure (**Fig. R6a**). This indicates the capacity of this system to overcome the intrinsic brittle nature of ceramics through structural engineering techniques, thereby expanding the application of 3D GQD@ZnO to wearable and flexible thermoelectric devices. Further supporting evidence is provided in **Fig. R6b, c**, demonstrating the flexibility of epoxy-filled 3D Bi_{1.5}Sb_{0.5}Te₃ and 3D Al₂O₃ fabricated through the PnP process. These findings suggest the feasibility of realizing future applications of 3D GQD@ZnO in flexible devices by incorporating a soft matrix into 3D GQD@ZnO.

From another practical perspective, 3D ZnO thermoelectric materials and the associated manufacturing technologies may be used for cooling applications in various electronic devices considering the thermoelectric effect of these systems. The fabrication procedure for 3D ZnO involves a photolithography-based proximity-field nanopatterning technique and atomic layer deposition process. These methodologies are extensively employed in the manufacturing of semiconductor devices. Therefore, 3D ZnO could be utilized for cooling applications in the electronic devices owing to its compatibility with current semiconductor processing technology. The relevant discussions have been added to the revised manuscript.

Fig. R6 | **a**, Finite element analysis (FEA) simulation of bulk ZnO and 3D thin-shell ZnO (*Nano Energy*, 2020, **78**, 105259 – our previous work). **b**, 3D Bi_{1.5}Sb_{0.5}Te₃/epoxy film transferred onto a flexible substrate (*J. Mater. Chem. C*, 2017, **5**, 8974-8980 – our previous work). **c**, Schematic and bending cycle test results of 3D alumina/epoxy-siloxane molecular hybrid (ESMH) nanocomposites (*Adv. Funct. Mater.*, 2021, **31**, 2010254 – our previous work).

Modification to the manuscript:

Page 4, added sentence:

Furthermore, the fabricated 3D ZnO has the potential for flexible thermoelectric applications, overcoming the intrinsic brittle nature of ZnO through structural engineering.³⁰

Page 14-15, added text:

To quantitatively demonstrate the beneficial effect of interfacing GQDs in 3D ZnO on the electronic transport properties, we calculated the weighted mobility for 3D GQD@ZnO (**Supplementary Fig. 12**). Weighted mobility is a direct indicator of the “electronic” performance of thermoelectric materials. Note that the weighted mobility is insensitive to the carrier concentration of a materials,⁴⁹ enabling a straightforward assessment of the GQD interfacing effect on the “electronic” performance. We observed a substantial enhancement in

the weighted mobility of ZnO upon GQD interfacing, indicating that GQDs effectively improve the electronic transport properties of 3D ZnO through an energy filtering effect.

Page 16, added sentence:

κ in the 3D GQD $_x$ @ZnO ($x = 4$) system also showed an opposite trend compared to that in the $x = 1, 2,$ and 3 systems (**Fig. 5b**), which is attributed to the increased MFP of phonons within the percolation network formed by GQD overloading (**Supplementary Fig. 16**).

Page 18, added and modified sentences:

Furthermore, we believe that the proposed approach can be extended to wearable and flexible thermoelectric devices, resulting from biocompatibility of the GQDs and unique mechanical properties of 3D ZnO systems. Finally, the lithography-based design of thermoelectric materials developed in this study has the potential for use in cooling applications in electronic devices, given its compatibility with current semiconductor processing technologies.

Page 25, added reference:

30. Kim H, *et al.* Breaking the elastic limit of piezoelectric ceramics using nanostructures: A case study using ZnO. *Nano Energy* **78**, 105259 (2020).

Page 26, added reference:

49. Snyder GJ, Snyder AH, Wood M, Gurunathan R, Snyder BH, Niu C. Weighted mobility. *Adv. Mater.* **32**, 2001537 (2020).

Page S-13, added supplementary figure:

Supplementary Figure 12| Temperature-dependent weighted mobility of 3D ZnO (small grain) and 3D GQD_x@ZnO ($x = 1, 2, 3,$ and 4).

Page S-17, added supplementary figure:

Supplementary Figure 16| Phonon mean free path (MFP) for bulk ZnO, 3D ZnO with different grain sizes, and 3D GQD_x@ZnO ($x = 1, 2, 3,$ and 4) at 300 K. Phonon MFP is calculated using kinetic theory based on the equation $k_{latt} = \frac{1}{3} C_v v l$, where k_{latt} is the lattice thermal conductivity, C_v is the specific heat capacity under a constant volume, v is the group velocity of phonons, and l is the phonon MFP.

Reviewer #2:

General Comments: *The authors have developed a nanostructured zinc oxide material as a thermoelectric material with low environmental load. Three steps were used as a strategy to improve the performance. First, the material itself is processed into a porous three-dimensional nano-periodic structure by using novel patterning technique to effectively scatter phonons with long MFPs and achieve lower thermal conductivity than the bulk ZnO. Next, thermal annealing was used to control the ZnO grain size, which was optimized to improve the electrical properties. Finally, nanoscale flack-shape graphene was modified on the surface to achieve high thermoelectric figure of merit zT by reducing thermal conductivity due to scattering of phonons with short MFPs and increasing Seebeck coefficient by filtering low energy electrons. It reports the lowest thermal conductivity and highest figure of merit zT compared to other nanostructured ZnO thermoelectric materials. It is worth mentioning that a very precise structural and composition analysis of the prepared samples was performed. The temperature dependent measurement of Seebeck coefficient and electrical conductivity, as well as the measurement of thermal conductivity and the estimation of its temperature dependence by theoretical calculation models, are also carefully performed and reasonable. Overall, I believe that publication could be considered by revising some of the expressions or, if possible, conducting additional experiments.*

Our response to General Comments: We thank the reviewer for these positive comments, and for the recommendation to publish in *Nature Communications*, and we have responded to all the reviewer's comments and modified the specified sections, as detailed below. We hope that these responses and modifications make the manuscript suitable for publication in *Nature Communications*.

Topic 1: *If high performance at high temperatures is to be strongly claimed, I believe that experimental measurement of thermal conductivity is essential. ($zT = 0.486$ and $\kappa = 0.785 \text{ W m}^{-1}\text{K}^{-1}$ at 580 K). If these values are claimed in abstract and conclusion, they should be mentioned as estimated values. If additional experiments are difficult, one option would be to focus on grain size control by annealing and the concentration dependence of graphene quantum dots.*

Our response to topic 1: We appreciate the reviewer's comment. Indeed, the direct measurement of thermal conductivity (κ) for relatively thin nano-porous structures is highly challenging. To address this issue, we used a microfabricated thermoelectric measurement platform (MTMP) to directly measure the in-plane thermal conductivity of the 3D ZnO and 3D GQD $_x$ @ZnO ($x = 1, 2, 3,$ and 4) system at 300 K based on Fourier's law. However, owing to the limitation of conducting κ measurements at high temperatures in the MTMP setup, a modified Debye-Callaway model based on the measured lattice κ at 300 K and electronic κ derived from the Wiedemann-Franz law was used to estimate the temperature-dependent κ of the 3D GQD@ZnO system. Following the reviewer's suggestions, we have investigated the additional effects of the grain size (d) in ZnO on the thermoelectric properties of this system to verify the extremely high thermoelectric performance achieved in this study. The grain size (d) of ZnO annealed at 250 °C shows a Gaussian-like distribution rather than a single value (**Fig. R1a**). Therefore, we have estimated the thermal conductivity for 3D GQD $_x$ @ZnO ($x = 3$), which shows the highest thermoelectric performance, within the range of $d = 10$ to 60 nm at 580 K. The estimation revealed a potential variation in the thermal conductivity from 0.681 to 0.879 $\text{W m}^{-1}\text{K}^{-1}$, corresponding to a variation in zT from 0.560 to 0.434 at 580 K (**Fig. R1b**). Note that the typical measurement error of zT is approximately 20% (*Nat. Mater.*, 2008, 7, 105-114). Furthermore, we indirectly calculated the density of GQDs acting as phonon scattering sites by measuring κ of the 3D ZnO and 3D GQD $_x$ @ZnO ($x = 1, 2,$ and 3) system at 300 K, corresponding to varying loading levels of GQDs (**Fig. R1c**). Note that the calculation of the GQD density in 3D GQD $_x$ @ZnO was performed until $x = 3$, as a percolation network of GQDs is formed at $x = 4$. In the calculation, GQDs were considered as nanoprecipitates, and the observed decreasing trend in κ with an increase in GQD loading levels implies an elevated density of GQDs, contributing significantly to effective phonon scattering. The relevant figure and discussions have been added to the revised manuscript.

Fig. R1| a, TEM image of the 3D ZnO system after annealing at 250 °C. The inset of (a) indicates the distribution of 2D plate-like grain sizes on the surface of thin-shell ZnO. **b**, Estimated thermal conductivity and zT values at 580 K for 3D GQD $_x$ @ZnO ($x = 3$) obtained using modified Debye-Callaway model as a function of the grain size. **c**, Measured thermal conductivity at 300 K and calculated density of GQDs for 3D GQD $_x$ @ZnO ($x = 1, 2$, and 3).

Modification to the manuscript:

Page 2, modified sentences in Abstract:

In this study, we used a multi-step strategy to achieve a significantly high dimensionless figure-of-merit (zT) value of approximately 0.486 at 580 K (estimated value) by interfacing graphene quantum dots (GQDs) with three-dimensional (3D) nanostructured ZnO (3D GQD@ZnO).

Page 2: modified sentences in Abstract:

Here, we show the fabrication of 3D GQD@ZnO with the highest zT value ever reported for ZnO counterparts; specifically, our experimental results indicate that the fabricated 3D GQD@ZnO exhibited a significantly low thermal conductivity of 0.785 W m⁻¹K⁻¹ (estimated value) and a remarkably high Seebeck coefficient of $-556 \mu\text{V K}^{-1}$ at 580 K.

Page 15, added sentences:

Please note that the estimated zT value for 3D GQD $_x$ @ZnO ($x = 3$) exhibits a potential variation from 0.434 to 0.560 at 580 K, corresponding to a variation in κ from 0.879 to 0.681 W m⁻¹K⁻¹ (Supplementary Fig. 15), considering the grain size distribution of ZnO within the range of 10 to 60 nm (Fig. 2g). It is noteworthy that the typical measurement error of zT is approximately 20%.⁵⁰

Page 18, modified sentences in Discussion:

The proposed 3D GQD@ZnO exhibited an extraordinarily high zT of 0.486 at 580 K (estimated value) owing to an ultralow thermal conductivity ($0.785 \text{ W m}^{-1}\text{K}^{-1}$ at 580 K, estimated value) and an extremely high Seebeck coefficient ($-556 \mu\text{V K}^{-1}$ at 580 K), outperforming all other ZnO-based materials that have been reported to date.

Page 26, added reference:

50. Snyder GJ, Toberer ES. Complex thermoelectric materials. *Nat. Mater.* **7**, 105-114 (2008).

Page S-16, added supplementary figure:

Supplementary Figure 15| Estimated thermal conductivity and zT values at 580 K for 3D GQD x @ZnO ($x = 3$) as a function of grain size for ZnO.

Topic 2: *I think it is unfair to discuss only at 580 K temperature in comparison with other materials. While the structure authors reported can break down at temperatures above 600 K, it is hard to consider using the material in exactly 580 K hot environment for actual applications. Or do you have any idea to improve temperature stability of GNDs@3D ZnO above 600 K?*

Our response to topic 2: Thank you for your constructive comment. As highlighted by the reviewer, the utilization of thermoelectric materials under the specific temperature condition of 580 K poses a challenge for practical thermoelectric systems. In our current study, we have interfaced GQDs on a 3D thin-shell ZnO nanostructure, where the GQDs are directly exposed to high temperatures. Consequently, the thermoelectric properties of 3D GQD@ZnO are abruptly reduced at temperatures exceeding 600 K due to the thermal decomposition of the GQDs (**Fig. R2a**). Therefore, we intentionally reported the limited zT value of 3D GQD@ZnO below 600 K to ensure the reliability of the thermoelectric performance of this system, as the zT value exhibiting the highest value yet reported for ZnO-based thermoelectric materials at the corresponding operating temperature (**Fig. R2b**). However, oxide-based thermoelectric materials are known to typically exhibit higher zT values at temperatures above 600 K. Following the reviewer's suggestion, we compared the thermoelectric performance of 3D GQD@ZnO with peak zT values previously reported for ZnO-based thermoelectric materials (**Fig. R2c**). As expected, ZnO shows high zT values at around 1000 K. This implies that 3D GQD@ZnO could potentially achieve higher zT value if the GQDs remain stable at higher temperatures. To address this issue, our follow-up study focuses on enhancing the thermal stability of 3D GQD@ZnO at higher temperatures by encapsulating the GQDs using thermally protective materials with superior thermal stability (**Fig. R2d**). In the field of optoelectronics, strategies for improving the thermal stability of quantum dots (QDs) involve encapsulating QDs between thermally protective materials, such as metal oxides, metal salts, organic polymers, and amorphous glass, *etc* (*Nano Lett.*, 2006, **6**, 800–808, *Angew. Chem., Int. Ed.*, 2016, **55**, 7924–7929, *Adv. Mater.*, 2017, **29**, 1701185, and *J. Mater. Chem. C*, 2019, **7**, 13585–13593). This encapsulation strategy may be used to advance the application of thermoelectric systems including QDs-based materials in high-temperature environments.

Fig. R2| a, Thermal gravimetric analysis (TGA) results of GQDs. zT values at 580 K (**b**) and peak zT values (**c**) of previously reported nanostructured ZnO-based thermoelectric materials. **d**, Strategy to improve the thermal stability of GQDs by encapsulating GQDs between thermally protective layers.

Topic 3: *In page 8, regarding the three-dimensional structure, the authors state that they have optimized the dimensions, but only one dimension (~70 nm) is reported in the reference 24 (10.1039/c7nr08167g) and the optimization needs to be explained. I believe that thermoelectric properties are strongly dependent on dimensions in this scale, especially thermal conductivity.*

Our response to topic 3: We apologize for the insufficient information. We fully agree with your comment. As noted by the reviewer, the dimensions of the 3D ZnO nanostructure, particularly the shell thickness of ZnO, can indeed influence phonon transport behavior. In fact, the total lattice thermal conductivity of materials results from the cumulative transport of phonons with various mean free paths (MFP). Therefore, we investigated the cumulative lattice thermal conductivity considering the phonon MFP for ZnO systems reported in the literature (*J. Mater. Chem. A*, 2020, **8**, 11948-11957). **Fig. R3a** shows the portion of the lattice thermal conductivity dependent on the phonon MFP for ZnO. Notably, phonons with MFPs below 100 nm make a 44% contribution of the overall thermal conductivity, while those in the 100-1000 nm range contribute 41%. Our 3D nanostructuring approach can selectively scatter the latter category of phonons. To investigate the optimal structural dimensions, we prepared three types of 3D ZnO with shell thicknesses of 30, 50, and 70 nm. The average pore sizes for these structures were 280, 230, and 180 nm, respectively, which depends on the shell thickness of ZnO (**Fig. R3b, c**). The reduction ratios of the thermal conductivity achieved by phonon scattering in 3D ZnO with different shell thicknesses are expected to be similar given the comparable thermal conductivity contributions of phonons with MFPs corresponding to each pore size (**Fig. R3a**). However, when the ZnO shell thickness exceeds 100 nm, the thermal conduction contributed by phonons with short MFPs, which are transported within the relatively thick shell (100 nm or greater), becomes more significant than the phonon scattering at the nanostructured surfaces. Meanwhile, as the shell thickness increases from 30 to 70 nm, the 3D ZnO becomes more uniform and exhibits high structural stability (**Fig. R3d**). For the 30-nm sample with a relatively thin-shell, the structural stability was relatively lower due to a slight collapse during the removal of the polymer template after ZnO deposition. The structural stability significantly impacts the reliability and reproducibility of the thermoelectric properties. Therefore, we optimized the shell thickness of 3D ZnO to 70 nm considering the factors of effective phonon scattering at nanostructured surfaces, structural stability, and the reliability of thermoelectric properties. The relevant figure and discussions have been added to the revised manuscript.

Fig. R3| Optimization of the shell thickness of 3D ZnO in terms of the thermal conductivity and structural stability. **a**, Cumulative lattice thermal conductivity against the phonon mean free path for ZnO (*J. Mater. Chem. A*, 2020, 8, 11948-11957). **b**, Schematic illustration of 3D thin-shell ZnO. **c**, Pore sizes of 3D ZnO with 30, 50, and 70 nm shell thickness. **d**, Cross-sectional view SEM images of 3D ZnO with 30, 50, and 70 nm shell thickness in terms of structural stability.

Modification to the manuscript:

Page 8, added sentences:

To optimize the shell thickness of the 3D ZnO with respect to phonon transport, we prepared three types of 3D ZnO systems with shell thicknesses of 30, 50, and 70 nm by adjusting the deposition cycles of the ALD process. The average pore sizes for three types of 3D ZnO systems were 280, 230, and 180 nm, respectively, which depend on the shell-thickness of ZnO (Supplementary Fig. 2a, b). The reduction ratios of the thermal conductivity achieved by phonon scattering in 3D ZnO with different shell thicknesses are expected to be similar, given the comparable contribution of thermal conductivity by phonons with MFPs corresponding to different pore sizes.⁴² However, when the ZnO shell thickness exceeds 100 nm, the thermal conduction achieved by phonons with a short MFP, which are transported within the relatively thick shell (100 nm or greater), becomes more significant than the phonon scattering at the

nanostructured surfaces.⁴² Meanwhile, as the shell thickness increases, the 3D ZnO nanostructure becomes more uniform and exhibits high structural stability (**Supplementary Fig. 2c**). For the 30-nm sample with a relatively thin shell, structural stability was relatively lower due to a slight collapse during the removal of the polymer template after ZnO deposition.

Page 26, added reference:

42. Spooner KB, Ganose AM, Scanlon DO. Assessing the limitations of transparent conducting oxides as thermoelectrics. *J. Mater. Chem. A* **8**, 11948-11957 (2020).

Page S-3, added supplementary figure:

Supplementary Figure 2 | Optimization of the shell thickness of 3D ZnO in terms of thermal conductivity and structural stability. **a**, Schematic illustration of 3D thin-shell ZnO. **b**, Pore sizes of 3D ZnO with 30, 50, and 70 nm shell thickness. **c**, Structural stability of 3D ZnO nanostructures according to the shell thickness. Error bars shown in the **Supplementary Fig. 2** represents the standard deviation (SD).

Topic 4: About the model for temperature dependence of thermal conductivity, rate for boundary scattering is defined as v/d , and 20 nm is used for d according to the supplemental information. Because this κ value is the basis for the main claim, I think it is inaccurate to use only one value. At least, it should have some range with in range of $d = 10\text{-}60$ nm according to the grain distribution for ZnO annealed at 250 degree.

Our response to topic 4: Thank you for your valuable comment. As pointed out by the reviewer, the grain size (d) of ZnO annealed at 250 °C shows a Gaussian-like distribution rather than a single value (**Fig. R4a**). Following the reviewer's suggestion, we have estimated the thermal conductivity for 3D GQD $_x$ @ZnO ($x = 3$), which exhibits the highest thermoelectric performance, within the range of $d = 10$ to 60 nm at 580 K. The estimation revealed a potential variation in thermal conductivity from 0.681 to 0.879 W m⁻¹K⁻¹, corresponding to a variation in zT from 0.560 to 0.434 at 580 K (**Fig. R4b**). Specifically, the estimated minimum zT value of 0.434 for 3D GQD $_x$ @ZnO ($x = 3$) with $d = 60$ nm surpasses the zT value of 0.1 observed for bare 3D ZnO without GQD interfacing. Note that the typical measurement error of zT is approximately 20% (*Nat. Mater.*, 2008, 7, 105-114). The relevant figure and discussions have been added to the revised manuscript.

Fig. R4| a, TEM image of 3D ZnO after annealing at 250 °C. The inset of (a) indicates the distribution of 2D plate-like grain sizes on the surface of thin-shell ZnO. **b**, Estimated thermal conductivity and zT values at 580 K for 3D GQD $_x$ @ZnO ($x = 3$) obtained using the modified Debye-Callaway model as a function of grain size.

Modification to the manuscript:

Page 15, added sentences:

Please note that the estimated zT value for 3D GQD x @ZnO ($x = 3$) exhibits a potential variation from 0.434 to 0.560 at 580 K, corresponding to a variation in κ from 0.879 to 0.681 $\text{W m}^{-1}\text{K}^{-1}$ (Supplementary Fig. 15), considering the grain size distribution of ZnO within the range of 10 to 60 nm (Fig. 2g). It is noteworthy that the typical measurement error of zT is approximately 20%.⁵⁰

Page 26, added reference:

50. Snyder GJ, Toberer ES. Complex thermoelectric materials. *Nat. Mater.* **7**, 105-114 (2008).

Page S-16, added supplementary figure:

Supplementary Figure 15| Estimated thermal conductivity and zT values at 580 K for 3D GQD x @ZnO ($x = 3$) as a function of grain size for ZnO.

Reviewer #3:

General Comments: *This manuscript reports a significantly high zT of 0.486 at 580 K in the 3D nanostructured ZnO with graphene quantum dots interfaced. The 3D nanostructured ZnO was fabricated by atomic-layer depositing the thin ZnO layer on the 3D nanostructured polymeric template. The structural design is very interesting and leads to a substantial reduction in κ without degrading σ , but it seems to have been reported many times (such as the references 24-26). In addition to the structural engineering, interfacing lowly-oxidized GQDs with the 3D ZnO support effectively allows for low-energy electron filtering and further phonon scattering at the interfacial energy barrier, which enhances S , and also leads to the significant reduction in κ from 1.49 to 0.785 $W m^{-1}K^{-1}$. Some concerns are listed below:*

Our response to General Comments: Thank you for your valuable comments on our work indicating that it is well presented. We agree with your constructive comments. We hope that our responses and modifications make the manuscript suitable for publication in *Nature Communications*.

Topic 1: Please show the repeatability of high thermoelectric properties of the 3D GQDs@ZnO 3 sample.

Our response to topic 1: We apologize for providing insufficient information. We conducted five sets of measurements with different samples for the thermoelectric properties of 3D GQD@ZnO to establish the reproducibility of its high thermoelectric performance. We verified the high thermoelectric performance for 3D GQD_x@ZnO ($x = 3$) with high reproducibility by conducting repeated measurements of its thermoelectric properties, which is attributed to the high structural uniformity of 3D ZnO and uniform dispersion of GQDs on the interface of ZnO. The relevant figure and discussions have been added to the revised manuscript.

Modification to the manuscript:

Page 15, added and modified sentences:

Based on the enhanced power factor and decreased κ resulting from structural and interface engineering techniques, the 3D GQD_x@ZnO ($x = 3$) exhibited the maximum zT of 0.486 at 580 K with a high reproducibility of the thermoelectric performance, which was determined by conducting repeated measurements of its thermoelectric properties (**Supplementary Fig. 14**).

Page 21, added and modified sentences:

The temperature-dependent σ and S values for all samples were measured five times with different samples to ensure their reliability.

Page 21, added sentence:

The thermal conductance (G) of 3D ZnO and 3D GQD@ZnO were measured under vacuum conditions (10^{-6} Torr) at 300 K. G values obtained via the MTMP system were converted into thermal conductivity (κ) considering the cross-sectional area (A) and length (l) of the samples using the following equation:

$$\kappa = \frac{G \cdot l}{A} \Delta T$$

κ for all samples was measured five times with different samples at 300 K to ensure its reliability.

Supplementary Figure 14| Five sets of measurements for the thermoelectric properties of 3D GQD@ZnO. Temperature dependent (a) Seebeck coefficient, (b) electrical conductivity, (c) power factor, (d) thermal conductivity, and (e) zT values of 3D GQD_x@ZnO ($x = 3$).

Topic 2: Discussion on the temperature-dependent carrier concentration and Hall mobility is needed.

Our response to topic 2: We appreciate this issue raised by the reviewer. First, we conducted Hall measurements at 300 K to assess charge carrier concentration (n) and charge carrier mobility (μ) and investigate the energy filtering effect at the GQD/ZnO heterogeneous interface (**Fig. R1a**). The results indicated that n gradually decreased in 3D GQD $_x$ @ZnO with an increase in the GQDs concentration from $x = 0$ to 3. The reduction in n with an increase in the GQD content was ascribed to the formation of an interfacial energy barrier of 0.62 eV at the GQD/ZnO heterogeneous interface, which acted as an energy trap for the conduction electrons within the 3D ZnO. In contrast, for the 3D GQD $_x$ @ZnO ($x = 4$), the n shows a different trend than that observed for $x = 1, 2,$ and 3. An abrupt increase of n in 3D GQD $_x$ @ZnO ($x = 4$) indicates that a conduction pathway of π -electrons generated from π - π stacking between GQDs is formed. Meanwhile, μ at 300 K was gradually decreased in 3D GQD $_x$ @ZnO with an increase in the GQD concentration from $x = 0$ to 4 owing to enhanced impurity scattering.

Second, we investigated the temperature dependence of n and μ within the temperature range of 300 to 580 K. A gradual increase in n with temperature was observed in 3D GQD $_x$ @ZnO ($x = 1, 2, 3,$ and 4) (**Fig. R1b**). This trend could be attributed to the presence of low-energy carriers, which are trapped at the interfacial energy barrier around 300 K, gradually overcoming this barrier due to thermal excitation. Notably, the release of trapped low-energy carriers with increasing temperature is generally observed in semiconducting thermoelectric materials (*J. Am. Chem. Soc.*, 2013, **135**, 7486-7495). The released low-energy carriers with increasing temperature are correlated with the temperature dependence trend of μ . A gradual decrease in μ with temperature was observed in 3D GQD $_x$ @ZnO ($x = 1, 2, 3,$ and 4) (**Fig. R1c**), suggesting enhanced electron-electron collisions with a gradual increase in n as temperature increases. The relevant figure and discussions have been added to the revised manuscript.

Fig. R1| a, Charge carrier concentration and charge carrier mobility of 3D ZnO (small grain) and 3D GQD x @ZnO ($x = 1, 2, 3,$ and 4) at 300 K. Temperature-dependent charge carrier concentration (**b**) and charge carrier mobility (**c**) of 3D ZnO (small grain) and 3D GQD x @ZnO ($x = 1, 2, 3,$ and 4).

Modification to the manuscript:

Page 14, added and modified sentences:

The Hall measurement obtained at 300 K confirmed that n gradually decreased in 3D GQD x @ZnO with an increase in the GQD concentration from $x = 0$ to 3 (Supplementary Fig. 11a). The reduction in n with an increase in the GQD content was ascribed to the formation of an interfacial energy barrier of 0.62 eV at the GQD/ZnO heterogeneous interface, which acted as an energy trap for the conduction electrons within the 3D ZnO.⁴⁸ Notably, the improvement in S through the energy filtering effect was generally observed when the potential energy barrier exceeded 0.3 eV, providing further validation of our results.³¹ As the temperature increased, the gradual increase in n was observed in 3D GQD@ZnO (Supplementary Fig. 11b). This trend could be attributed to low-energy carriers, which are trapped at the interfacial energy barrier around 300 K, gradually overcoming the barrier due to thermal excitation.⁴⁸ Meanwhile, charge carrier mobility (μ) at 300 K showed a gradual decrease in 3D GQD@ZnO with an increase in the GQD concentration (Supplementary Fig. 11a), which is attributed to enhanced impurity scattering. As the temperature increased, μ gradually decreased in all 3D GQD@ZnO systems due to the enhanced electron-electron collisions with a gradual increase in n (Supplementary Fig. 11c).

Meanwhile, the σ value of the 3D GQD $_x$ @ZnO gradually decreased from 2531 S m $^{-1}$ (for $x = 0$) to 2123 S m $^{-1}$ (for $x = 3$) at 580 K (**Supplementary Fig. 11d**), indicating charge carrier scattering at numerous heterogeneous interfaces (**Supplementary Fig. 11a**). However, the increase in S (44%) in the interfacing GQDs (from $x = 0$ to 3) was significantly greater than the reduction in σ (19%), indicating that high-energy electrons contribute more to the overall electrical conduction than low-energy electrons. The resultant power factor of the 3D GQD $_x$ @ZnO system gradually increased from 0.377 mW m $^{-1}$ K $^{-2}$ (for $x = 0$) to 0.657 mW m $^{-1}$ K $^{-2}$ (for $x = 3$) at 580 K, representing a 74% performance improvement in interfacing GQDs (**Supplementary Fig. 11e**).

Page S-12, added and modified supplementary figure:

Supplementary Figure 11 | Hall measurement results of 3D GQD@ZnO. a, Charge carrier concentration and charge carrier mobility of 3D ZnO (small grain) and 3D GQD $_x$ @ZnO ($x = 1, 2, 3,$ and 4) at 300 K. **b,** Temperature-dependent charge carrier concentration and **c,** charge carrier mobility of the 3D ZnO (small grain) and 3D GQD $_x$ @ZnO ($x = 1, 2, 3,$ and 4). **d,** Temperature-dependent electrical conductivity and **(e)** power factor for 3D ZnO (small grain) and 3D GQD $_x$ @ZnO ($x = 1, 2, 3,$ and 4). Error bars shown in the **Supplementary Fig. 11** represents the SD.

Topic 3: Please show the thermal gravimetric analysis result for the sample of 3D GQDs@ZnO 3 within the temperature range of 300–580 K.

Our response to topic 3: We apologize for providing insufficient information. As requested by the reviewer, we conducted thermal gravimetric analysis (TGA) on the 3D GQD_x@ZnO ($x = 3$) system (Fig. R2a). No significant weight loss was observed in 3D GQD_x@ZnO ($x = 3$) due to the weight difference between 3D ZnO and the interfaced GQDs. Nevertheless, the TGA results for the bare GQDs clearly revealed their decomposition starting around 600 K. Consequently, the thermoelectric properties of 3D GQD_x@ZnO ($x = 3$) are abruptly reduced at temperatures exceeding 600 K due to the thermal decomposition of the GQDs. To address this issue, our follow-up study focuses on enhancing the thermal stability of 3D GQD@ZnO at higher temperatures by encapsulating the GQDs using thermally protective materials with superior thermal stability (Fig. R2b). The relevant figure and discussions have been added to the revised manuscript.

Fig. R2| a, Thermal gravimetric analysis (TGA) results of bare GQDs and 3D GQD@ZnO. **b,** Strategy to the improve thermal stability of GQDs by encapsulating GQDs between thermally protective layers.

Modification to the manuscript:

Page 15: added and modified sentences:

It should be noted that we evaluated the thermoelectric parameters within the temperature range of 300–580 K, considering the thermal stability of GQDs. The temperature-dependent weight loss in 3D GQD@ZnO, as evaluated by thermal gravimetric analysis (TGA), was negligible due to the weight difference between the 3D ZnO and interfaced GQDs. However, the TGA results clearly showed the thermal decomposition of GQDs around 600 K (Supplementary Fig. 13a). Consequently, temperatures beyond 600 K exhibit an abrupt decrease in S owing to the thermal decomposition of GQDs (Supplementary Fig. 13b).

Supplementary Figure 13 | Thermal stability of GQDs. **a**, Thermal gravimetric analysis (TGA) results of bare GQDs and 3D GQD@ZnO. **b**, Seebeck coefficient of 3D ZnO (small grain) and 3D GQD_x@ZnO ($x = 3$) within the temperature range of 300–670 K. Error bars shown in the **Supplementary Fig. 13** represents the SD.

Topic 4: *Since the highest zT achieved in this work outperforms any other ZnO-based materials reported to date, it is necessary to show the κ measurement result at high temperatures rather than estimated data.*

Our response to topic 4: We appreciate the reviewer's comment. The main aim of this work is to fabricate a novel 3D nanostructure with a high zT value and experimentally demonstrate its performance. Unlike electric current flow, heat can flow in any material, leading to the possibility of heat leakage or parasitic heat flow, which can result in significant measurement uncertainties. This challenge becomes more pronounced in nanomaterials with a small heat capacity, where even a minimal amount of thermal leakage or parasitic heat flow can significantly impact the results. Therefore, selecting a reliable method for measuring thermal conductivity (κ) is crucial, and we combined an experimental method with a theoretical estimation approach. We used MEMS-based steady-state measurements offering precise and reliable data and a modified Debye-Callaway model providing a theoretical framework suited for considering various scatterings at interfaces at low and high temperatures.

In this work, we used a microfabricated thermoelectric measurement platform (MTMP) to directly measure the in-plane κ of 3D ZnO and 3D GQD $_x$ @ZnO ($x = 1, 2, 3,$ and 4) at 300 K based on Fourier's law. However, there is a limitation of conducting direct κ measurements using the MTMP setup at high temperatures where the maximum zT value was observed. This is due to the risk of distortion in our finely processed membrane structures of the MTMP and potential damage to electrical wiring at high temperatures. Additional insulation is needed to handle such high temperatures, but it may reduce measurement sensitivity, making precise measurements difficult to achieve. As an alternative to directly measuring the κ of 3D GQD@ZnO at high temperatures, we employed the Debye-Callaway model, which is widely used for estimating lattice κ , to predict κ at high temperatures. This model, based on the relaxation time approximation, has been used to successfully investigate the effects of various scatterings resulting from microstructural modifications. The relaxation time for each process is given as follows:

$$\tau_U^{-1} + \tau_N^{-1} = \beta \frac{2}{(6\pi^2)^{1/3}} \frac{k_B \bar{v}^{1/3} \gamma^2 \omega^2 T}{\bar{M} v^3} \quad (\text{Umklapp and normal process})$$

$$\tau_B^{-1} = \frac{v}{d} \quad (\text{Boundary scattering based on grain boundary engineering of the 3D ZnO})$$

$$\tau_l^{-1} = v_s N_p \left[\left((2\pi R^2)^{-1} + \left(\frac{4}{9} \pi R^2 \left(\frac{\Delta D}{D} \right)^2 \left(\frac{\omega R}{v_s} \right)^4 \right)^{-1} \right) \right]^{-1} \quad \text{(Nanoprecipitate scattering based on GQD interfacing in 3D ZnO)}$$

The total relaxation time is given by Matthiessen's rule as $\tau_{tot}^{-1}(x) = \tau_U^{-1}(x) + \tau_N^{-1}(x) + \tau_B^{-1} + \tau_l^{-1}$. Finally, the effect of porosity based on 3D nanostructuring was considered by the effective medium theory, $\kappa_{l,porous} = (1 - P)^{3/2} \kappa_{l,dense}$, where P is the porosity of the given material. This estimation comprehensively involves complex variables by considering the material parameters of the 3D GQD@ZnO system developed in this study. Furthermore, we believe that it is a well-established calculation, as it has been employed in several thermoelectric studies to estimate the high temperature κ of thermoelectric materials (*Nanoscale*, 2018, **10**, 3046-3052 and *ACS Energy Lett.*, 2022, **7**, 2092-2101).

To further ensure the reliability of estimated κ values and verify the extremely high thermoelectric performance of the investigated system, we studied the additional effect of grain size (d) for ZnO. d of 3D ZnO annealed at 250 °C exhibits a Gaussian-like distribution rather than a single value (Fig. R3a). Consequently, we estimated κ for 3D GQD x @ZnO ($x = 3$), which exhibits the highest thermoelectric performance, within the range of $d = 10$ to 60 nm at 580 K. The estimation revealed a potential variation in κ from 0.681 to 0.879 W m⁻¹K⁻¹, corresponding to a variation in zT from 0.560 to 0.434 at 580 K (Fig. R3b). It is important to note that the typical measurement error of zT is approximately 20% (*Nat. Mater.*, 2008, **7**, 105-114). The relevant figure and discussions have been added to the revised manuscript.

Fig. R3| a, TEM image of the 3D ZnO after annealing at 250 °C. The inset of (a) indicates the

distribution of 2D plate-like grain sizes on the surface of thin-shell ZnO. **b**, Estimated thermal conductivity and zT values at 580 K for 3D GQD $_x$ @ZnO ($x = 3$) obtained using the modified Debye-Callaway model as a function of the grain size.

Modification to the manuscript:

Page 15, added sentences:

Please note that the estimated zT value for 3D GQD $_x$ @ZnO ($x = 3$) exhibits a potential variation from 0.434 to 0.560 at 580 K, corresponding to a variation in κ from 0.879 to 0.681 $\text{W m}^{-1}\text{K}^{-1}$ (Supplementary Fig. 15), considering the grain size distribution of ZnO within the range of 10 to 60 nm (Fig. 2g). It is noteworthy that the typical measurement error of zT is approximately 20%.⁵⁰

Page 26, added reference:

50. Snyder GJ, Toberer ES. Complex thermoelectric materials. *Nat. Mater.* **7**, 105-114 (2008).

Page S-16, added supplementary figure:

Supplementary Figure 15| Estimated thermal conductivity and zT values at 580 K for 3D GQD $_x$ @ZnO ($x = 3$) as a function of grain size for ZnO.

Additional part:

The manuscript has been proofread and corrected by a professional English editing company, and modifications have been updated in the revised manuscript.

The terminology defining the heterostructure comprising QDs and 3D ZnO, with varying QD loading levels, has been modified from “3D QDs@ZnO_x ($x = 1, 2, 3,$ and 4)” to “3D QD_x@ZnO ($x = 1, 2, 3,$ and 4)” to avoid potential confusion considering the oxidation degree of ZnO.

The title has been modified for correction.

Modification to the manuscript:

Page 1, modified Title:

High Figure-of-merit for ZnO Nanostructure by Interfacing Lowely-Oxidized Graphene Quantum dots (original title)

High Figure-of-merit for ZnO Nanostructures by Interfacing Lowely-Oxidized Graphene Quantum dots (corrected title)

The abstract has been modified to comply with the word count guideline of the *Nature Communications* journal.

Modification to the manuscript:

Page 2, modified Abstract:

Thermoelectric technology has potential for converting waste heat into electricity. Although traditional thermoelectric materials exhibit extremely high thermoelectric performances, their scarcity and toxicity limit their applications. Zinc oxide (ZnO) is a potential thermoelectric material owing to its high thermal stability and relatively high Seebeck coefficient, while also being earth-abundant and nontoxic. However, its high thermal conductivity ($> 40 \text{ W m}^{-1}\text{K}^{-1}$) remains a

challenge. In this study, we used a multi-step strategy to achieve a significantly high dimensionless figure-of-merit (zT) value of approximately 0.486 at 580 K (estimated value) by interfacing graphene quantum dots (GQDs) with three-dimensional (3D) nanostructured ZnO (3D GQD@ZnO). Here, we show the fabrication of 3D GQD@ZnO with the highest zT value ever reported for ZnO counterparts; specifically, our experimental results indicate that the fabricated 3D GQD@ZnO exhibited a significantly low thermal conductivity of $0.785 \text{ W m}^{-1} \text{ K}^{-1}$ (estimated value) and a remarkably high Seebeck coefficient of $-556 \mu\text{V K}^{-1}$ at 580 K.

The reference has been replaced for correction.

Modification to the manuscript:

Page 26, replaced reference:

48. Han L. High temperature thermoelectric properties of ZnO based materials. dissertation for PhD/Linderorth Søren.–Lyngby-Taarbæk, 2014.–106 p (2014). (original reference)
55. Han L, *et al.* Scandium-doped zinc cadmium oxide as a new stable n-type oxide thermoelectric material. *J. Mater. Chem. A* **4**, 12221-12231 (2016). (replaced reference)

The section of ‘Authors and Affiliations’ has been added.

Modification to the manuscript:

Page 28-29, added section of ‘Authors and Affiliations’:

Authors and Affiliations

Department of Materials Science and Engineering, Korea University, Seoul 02841, Republic of Korea

Myungwoo Choi & Seokwoo Jeon

Department of Materials Science and Engineering, Korea Advanced Institute of Science and Technology (KAIST), Daejeon 34141, Republic of Korea

Juyoung An, Hanhwi Jang & Ji Hong Park

Interdisciplinary Materials Measurement Institute, Korea Research Institute of Standards and Science (KRISS), Daejeon 34141, Republic of Korea

Hyejeong Lee & Hosun Shin

Institute of Advanced Composite Materials, Korea Institute of Science and Technology (KIST), Jeonbuk 55324, Republic of Korea

Ji Hong Park & Seung Min Kim

Thin Film Materials Research Center, Korea Research Institute of Chemical Technology (KRICT), Daejeon 34114, Republic of Korea

Donghwi Cho

Department of Semiconductor Engineering, Pohang University of Science and Technology (POSTECH), Pohang 37673, Republic of Korea

Jae Yong Song

Department of Materials Science and Engineering, Hanbat National University, Daejeon 34158, Republic of Korea

Min-Wook Oh

The section of 'Rights and permissions' has been added.

Modification to the manuscript:

Page 30, added section of 'Rights and permissions':

Rights and permissions

Open Access This article is licensed under a Creative Commons Attribution 4.0 International License, which permits use, sharing, adaptation, distribution and reproduction in any medium or format, as long as you give appropriate credit to the original author(s) and the source, provide a link to the Creative Commons license, and indicate if changes were made. The images or other third party material in this article are included in the article's Creative Commons license, unless indicated otherwise in a credit line to the material. If material is not included in the article's Creative Commons license and your intended use is not permitted by statutory regulation or exceeds the permitted use, you will need to obtain permission directly from the copyright holder. To view a copy of this license, visit

<http://creativecommons.org/licenses/by/4.0/>.

A sentence defining the error bar has been added.

Modification to the manuscript:

Page 31, added sentence:

Error bars shown in the **Fig. 1** represents the standard deviation (SD).

Page 33, added sentence:

Error bars shown in the **Fig. 3** represents the SD.

Page 35, added sentence:

Error bars shown in the **Fig. 5** represents the SD.

Page S-3, added sentence:

Error bars shown in the **Supplementary Fig. 2** represents the standard deviation (SD).

Page S-12, added sentence:

Error bars shown in the **Supplementary Fig. 11** represents the SD.

Page S-14, added sentence:

Error bars shown in the **Supplementary Fig. 13** represents the SD.

REVIEWER COMMENTS

Reviewer #2 (Remarks to the Author):

Thank you for your response. The manuscript has been appropriately revised in response to the comments of the reviewers.

I would like to point out one more thing about Figure 5 and its caption. In Figure 5d, I would like to suggest to use filled star and open stars in red color for points of 3D GQD@ZnO to indicate that high temperature results contain the estimation of thermal conductivities, while using other color for points of 3D ZnO w/o GQD. In Figure 5e, how about plot with Temperature for x-axis in range of 200 K to 1200 K and ZT for y-axis in range of 0.0 to 0.6.

Reviewer #3 (Remarks to the Author):

The manuscript has been well revised. I think the revised manuscript is worthy of publication in Nature Communications.

<Responses to the reviewers' comments>

Manuscript ID: NCOMMS-23-41901A

Title: High Figure-of-merit for ZnO Nanostructures by Interfacing Lowly-Oxidized Graphene Quantum Dots

We appreciate the valuable comments from the reviewer. We have revised the manuscript according to the reviewer's directions. Thank you very much.

REVIEWER REPORT(S):

Reviewer #2:

General Comments: *Thank you for your response. The manuscript has been appropriately revised in response to the comments of the reviewers.*

Our response to General Comments: We appreciate the reviewer for the valuable comments and for the recommendation to publish in *Nature Communications*. We have responded to the reviewer's comments and modified the specified sections, as detailed below. We hope that these responses and modifications make the manuscript suitable for publication in *Nature Communications*.

Topic 1: *I would like to point out one more thing about Figure 5 and its caption. In Figure 5d, I would like to suggest to use filled star and open stars in red color for points of 3D GOD@ZnO to indicate that high temperature results contain the estimation of thermal conductivities, while using other color for points of 3D ZnO w/o GOD. In Figure 5e, how about plot with Temperature for x-axis in range of 200 K to 1200 K and ZT for y-axis in range of 0.0 to 0.6.*

Our response to topic 1: Thank you for your valuable comment. Following the reviewer's suggestion, we have modified the **Figure 5d, e**. The relevant figure and discussions have been added to the revised manuscript.

Modification to the manuscript:

With the significant reduction in κ , improved power factor attributed to the effective low-energy electrons filtering at numerous heterogeneous interfaces results in the highest zT record at 580K for nanostructured ZnO-based thermoelectric materials to the best of our knowledge (Fig. 5e).^{14, 27, 52, 53, 54, 55, 56, 61, 62} It should be note that we intentionally showed the limited zT value of 3D GQD@ZnO below 600 K to ensure the reliability of the thermoelectric performance of this system, considering the thermal stability of GQDs (Supplementary Fig. 13a, b).

The proposed 3D GQD@ZnO exhibited an extraordinarily high zT of 0.486 (estimated value) at 580 K owing to an ultralow thermal conductivity ($0.785 \text{ W m}^{-1}\text{K}^{-1}$ at 580 K, estimated value) and an extremely high Seebeck coefficient ($-556 \mu\text{V K}^{-1}$ at 580 K), outperforming all other ZnO-based materials within the temperature range below 600 K.

Fig. 5| Impact of GQDs on thermoelectric performance of 3D ZnO. Temperature-dependent (a) Seebeck coefficient, (b) thermal conductivity, and (c) zT of 3D ZnO (small grain) and 3D GQD_x@ZnO ($x = 1, 2, 3,$ and

4). Filled circles are measured values and open circles are estimated values. (d) Comparison of previously reported non-nanostructuring and nanostructuring approaches to thermal conductivity of ZnO as a function of temperature. Filled stars are measured value and open stars are estimated values. (e) Temperature-dependent zT values of previously reported nanostructured ZnO-based thermoelectric materials. Error bars shown in the Fig. 5 represents the SD.

Reviewer #3:

General Comments: *The manuscript has been well revised. I think the revised manuscript is worthy of publication in Nature Communications.*

Our response to General Comments: We thank the reviewer for the recommendation to publish in *Nature Communications*.

REVIEWERS' COMMENTS

Reviewer #2 (Remarks to the Author):

The manuscript has been well revised. I think the results of the experimental and theoretical calculations are fairly reported and will be of interest to many readers of multidisciplinary journal.

<Responses to the reviewers' comments>

Manuscript ID: NCOMMS-23-41901B

Title: High Figure-of-merit for ZnO Nanostructures by Interfacing Lowly-Oxidized Graphene Quantum Dots

REVIEWER REPORT(S):

Reviewer #2:

General Comments: *The manuscript has been well revised. I think the results of the experimental and theoretical calculations are fairly reported and will be of interest to many readers of multidisciplinary journal.*

Our response to General Comments: *We appreciate the reviewer for the recommendation to publish in *Nature Communications*.*